# POSITIONAL ENCODING FOR SPIKING TRANSFORMERS

## ABSTRACT

Spiking Neural Networks (SNNs) offer superior energy efficiency compared to Artificial Neural Networks (ANNs). Recent Transformer-based SNNs have achieved promising performance by integrating spike-driven computation with Transformer architectures. Positional information is essential in sequential tasks. However, existing positional encoding methods designed for ANNs cannot be directly applied to SNNs, as they interfere with the spike-driven computation paradigm, highlighting the need for SNN-specific solutions. We propose Spiking Positional Encoding (SPE), a novel positional encoding specifically designed for Spiking Transformers that captures both absolute and relative positional information. Its key component is the Positional Encoding Leaky Integrate-and-Fire (PE-LIF) neuron layer, which encodes positional information directly into neuron thresholds. Through continuous spike firing and membrane potential reset processes, this positional information is effectively reflected in the emitted spikes while preserving the spike-driven computation paradigm. Comprehensive experiments across seven datasets, including three time-series forecasting tasks and four natural language processing benchmarks, demonstrate that SPE consistently outperforms existing positional encoding methods and achieves state-of-the-art performance. SPE provides a tailored positional encoding solution for Spiking Transformers, bridging the performance gap between ANNs and SNNs, thus advancing neuromorphic computing applications in sequential modeling tasks.

## 1 INTRODUCTION

Spiking Neural Networks (SNNs), characterized as third-generation artificial neural networks, have attracted considerable research interest due to their superior energy efficiency and enhanced biological plausibility compared to conventional artificial neural network (ANN) architectures (Maass, 1997). Unlike conventional ANNs that rely on continuous-valued activations, SNNs employ binary spike events as information carriers and operate within a temporally sparse, event-driven processing paradigm (Zhang et al., 2021). The inherently sparse synaptic communication in SNNs enables the replacement of computationally intensive multiply-accumulate (MAC) operations with more efficient accumulate (AC) operations, resulting in substantial improvements in computational efficiency (Li et al., 2023; Xu et al., 2024). This energy-efficient property has driven the development of specialized neuromorphic hardware implementations, including SpiNNaker (Painkras et al., 2013), TrueNorth (Akopyan et al., 2015), Loihi (Davies et al., 2018), and Tianjic (Pei et al., 2019).

Despite these advantages, existing SNNs typically suffer from performance limitations that constrain their widespread application, particularly when compared to their ANN counterparts. To address these performance gaps while preserving the inherent benefits of spike-driven computation, recent years have witnessed several studies integrating Transformers into SNNs, leading to a series of high-performance models, such as Spikformer (Zhou et al., 2023b), Spikingformer (Zhou et al., 2023a), Spike-Driven Transformer v1, v2, v3 (Yao et al., 2024b;a; 2025), QKformer (Zhou et al., 2024) and SpikeLM (Xing et al., 2024). Compared to traditional convolutional architectures in SNNs, these Transformer-based models have demonstrated significant performance improvements (Zhang et al., 2022), highlighting their potential to integrate the computational efficiency of SNNs with the powerful expressive capability of Transformer architectures.

In sequential tasks such as time-series forecasting and natural language processing, positional information is crucial. However, Transformer models suffer from an inherent limitation of being position-agnostic, thus lacking the capability to capture positional information. To address this issue, positional encoding techniques have been proposed. While numerous positional encoding methods exist in the ANN domain (Vaswani et al., 2017; Shaw et al., 2018; Devlin et al., 2019; Lan et al., 2019; Su et al., 2024), these approaches cannot be directly applied to SNNs as they would disrupt the spike-driven paradigm. Positional encoding methods are primarily categorized into two types: absolute and relative positional encoding. Absolute positional encoding provides global positional information with advantages of low computational complexity and simple implementation. However, it can not represent relative positional information between different tokens, which is essential for capturing time-translational invariance property in sequential modeling. Relative positional encoding, on the other hand, captures relative positional information effectively but often struggles to represent global positional information and is typically incompatible with linear attention mechanisms. Therefore, developing a positional encoding method tailored for Spiking Transformers that combines the advantages of both absolute and relative positional encoding is necessary.

In this paper, we first analyze why Spiking Transformers require positional encoding, then propose four design principles for positional encoding method tailored to Spiking Transformers. Guided by these principles, we introduce Spiking Positional Encoding (SPE), a method specifically designed for Spiking Transformers that encodes both absolute and relative positional information. The core component of SPE is the Positional Encoding Leaky Integrate-and-Fire (PE-LIF) neuron layer, a variant of the soft-reset LIF neuron, which encodes positional information directly into neuron thresholds. This information is then reflected in the emitted spikes through continuous spike firing and membrane potential reset processes. We theoretically demonstrate that SPE possesses the capability to represent relative positional information and prove that SPE exhibits desirable long-term decay properties. Notably, SPE fully satisfies the four design principles we established, confirming the soundness of its design. Experimentally, we evaluate SPE on three time-series forecasting datasets and four natural language processing datasets, achieving state-of-the-art performance.

- **Problem Analysis and Design Principles.** We present an analysis of the necessity of positional encoding in Spiking Transformers and establish four design principles that guide the development of effective positional encoding methods tailored for Spiking Transformers.

- **Novel Positional Encoding Method for SNNs.** We propose SPE, a novel positional encoding approach specifically designed for Spiking Transformers that captures both absolute and relative positional information through the innovative PE-LIF neuron layer, which encodes positional information directly into neuron thresholds.

- **Theoretical Validation of SPE Properties.** We provide rigorous theoretical analysis demonstrating that SPE possesses the capability to represent relative positional information and prove that it exhibits desirable long-term decay properties, ensuring adherence to our proposed four design principles.

- **Comprehensive Experimental Validation.** We conduct extensive experiments across seven diverse datasets spanning three time-series forecasting tasks and four natural language processing benchmarks, demonstrating that SPE achieves state-of-the-art performance compared to existing positional encoding methods.

## 2 RELATED WORKS

### 2.1 SPIKING TRANSFORMERS

Spiking Transformers integrate SNNs with Transformer architectures, combining biological plausibility and energy efficiency with powerful representation capabilities. Spikformer (Zhou et al., 2023b) pioneered this integration with Spiking Self Attention (SSA), achieving the first spike-driven computation in Transformers. Spike-driven Transformer (Yao et al., 2023) advanced this concept with a linear-complexity self-attention mechanism that significantly reduces energy requirements. This architecture was later expanded into Spike-driven Transformer V2 (Yao et al., 2024a) to enhance versatility across vision tasks. Spike-driven Transformer V3 (Yao et al., 2025) addressed inherent limitations in spiking neurons through Spike Firing Approximation, enabling integer training and spike-driven inference while improving accuracy and efficiency. Beyond the aforemen-

tioned models, numerous excellent Spiking Transformers exist in the vision domain, including QK-former (Zhou et al., 2024), SNN-ViT (Wang et al., 2025), and $\alpha$-SSA-ViT (Xiao et al., 2025). For language modeling, SpikeLM (Xing et al., 2024) presented the first fully spiking mechanism for general language tasks with bi-directional and elastic spike encoding, narrowing the performance gap between SNNs and ANNs. Despite sophisticated model designs in these works, positional encoding remains largely unexplored. Convolutional layers prevalent in vision Spiking Transformers can serve as implicit positional encoding due to their inherent positional bias. However, this approach is unsuitable for sequence tasks. Sequence-oriented models like SpikeLM also lack specialized positional encoding designs. Therefore, developing dedicated positional encoding techniques for Spiking Transformers is essential.

## 2.2 Positional Encoding

In sequence tasks, sequential information is crucial, yet Transformers exhibit limited capability in capturing sequential information, necessitating the incorporation of positional encoding to address this deficiency. Positional encoding primarily comprises two categories: absolute positional encoding and relative positional encoding. Absolute positional encoding directly integrates positional information into the input. The original absolute positional encoding was generated through a predefined function (Vaswani et al., 2017), subsequently followed by learnable absolute positional encoding (Devlin et al., 2019; Lan et al., 2019). Since absolute positional encoding cannot capture relative positional information between different tokens, relative positional encoding was proposed. Relative positional encoding methods typically encode the relative position information into the attention mechanism. For instance, in (Shaw et al., 2018), trainable parameters representing relative positions are incorporated when computing attention scores and weighted values. RoPE (Su et al., 2024) combines the advantages of absolute positional encoding (implementation simplicity and computational efficiency) with the benefits of relative positional encoding (capturing relative positional information), and has become the mainstream choice for various LLMs. The aforementioned positional encoding methods from the ANN domain cannot be directly applied to SNNs, as they would compromise the spike-driven characteristics of SNNs. Existing positional encoding methods in the SNN domain, such as CPG-PE (Lv et al., 2024a), belong to absolute positional encoding and cannot represent relative positional information between tokens. Therefore, developing a positional encoding method for Spiking Transformers that combines the advantages of both absolute and relative positional encoding is necessary.

## 3 Preliminaries

### 3.1 Leaky Integrate-and-Fire Neuron

SNNs rely on spiking neurons as their fundamental processing units. These models aim to replicate the information processing capabilities of biological neurons. Several prominent examples include the Hodgkin-Huxley (Hodgkin & Huxley, 1952), Izhikevich (Izhikevich, 2003), and Leaky Integrate-and-Fire (LIF) neurons (Wu et al., 2018). Due to its computational efficiency, the LIF model is widely adopted in Spiking Transformers and has become the most mainstream neuron model. Its membrane potential, a key element of a neuron's firing behavior, is mathematically described as,

$$\hat{u}_{i,j}(t+1) = \tau u_{i,j}(t) + I_{i,j}(t), i = 1, 2, ..., N, j = 1, 2, ..., D, \tag{1}$$

where $u_{i,j}(t)$ and $I_{i,j}(t)$ represent the membrane potential and pre-synaptic input at time $t$ of the neuron in the $i$-th row and $j$-th column of the neuron layer, respectively, and $\hat{u}_{i,j}(t)$ is the intermediate representation of $u_{i,j}(t)$. $\tau$ is the constant leaky factor. The size of the neuron layer is $N \times D$, where $N$ represents the number of tokens and $D$ represents the dimension of vectors. When the membrane potential exceeds the threshold, the neuron fires a spike. Therefore, given the threshold $\theta_{i,j}$ of the neuron in the $i$-th row and $j$-th column, the firing function can be described as,

$$s_{i,j}(t+1) = \Theta\left(\hat{u}_{i,j}(t+1) - \theta_{i,j}\right), i = 1, 2, ..., N, j = 1, 2, ..., D, \tag{2}$$

where $\Theta$ represents the Heaviside step function and $s_{i,j}(t)$ represents the output spike. After spike firing, the membrane potential will be reset. Common reset methods are divided into two types: hard reset and soft reset, which can be described by Equation 3 and Equation 4, respectively.

$$u_{i,j}(t+1) = [1 - s_{i,j}(t+1)]\,\hat{u}_{i,j}(t+1), i = 1, 2, ..., N, j = 1, 2, ..., D, \tag{3}$$

$$u_{i,j}(t+1) = \hat{u}_{i,j}(t+1) - s_{i,j}(t+1) \cdot \theta_{i,j}, i = 1, 2, ..., N, j = 1, 2, ..., D. \tag{4}$$

## 3.2 Spiking Self-Attention

In Spikformer, a spiking version of self-attention named SSA is proposed, which is more suitable for SNNs than vanilla self-attention. Given an input feature sequence $X \in \mathbb{R}^{T \times N \times D}$, the SSA has three key components, namely query ($Q$), key ($K$), and value ($V$) which are calculated by learnable linear matrices $W_Q, W_K, W_V \in \mathbb{R}^{D \times D}$ and $X$:

$$Q = \mathcal{SN}_Q(\text{BN}(XW_Q)), K = \mathcal{SN}_K(\text{BN}(XW_K)), V = \mathcal{SN}_V(\text{BN}(XW_V)), \tag{5}$$

where $\mathcal{SN}$ is a spike neuron layer described in subsection 3.1 and $T$ is the number of time steps. BN represents batch normalization. The output of SSA can be computed as:

$$\text{SSA}'(Q, K, V) = \mathcal{SN}(QK^\top V \cdot \sigma), \tag{6}$$

$$\text{SSA}(Q, K, V) = \mathcal{SN}(\text{BN}(\text{Linear}(\text{SSA}'(Q, K, V)))), \tag{7}$$

where $\sigma$ is a scaling factor to control the large values of the matrix multiplication results. $QK^\top$ is referred to as the attention map.

A key advantage of SSA lies in its linear attention property, whereby the time complexity of SSA can be approximated as $\mathcal{O}(N)$ when the number of tokens $N$ significantly exceeds the token dimension $D$. This is attributable to the absence of Softmax operations in SSA, which enables the computational order of $QK^\top V$ to be restructured from computing $QK^\top$ first to computing $K^\top V$ first. Consequently, the computational complexity is changed from $\mathcal{O}(N^2D)$ to $\mathcal{O}(ND^2)$. When $N \gg D$, this yields $\mathcal{O}(ND^2) \sim \mathcal{O}(N)$.

## 4 Method

### 4.1 Problem Analysis and Design Principles

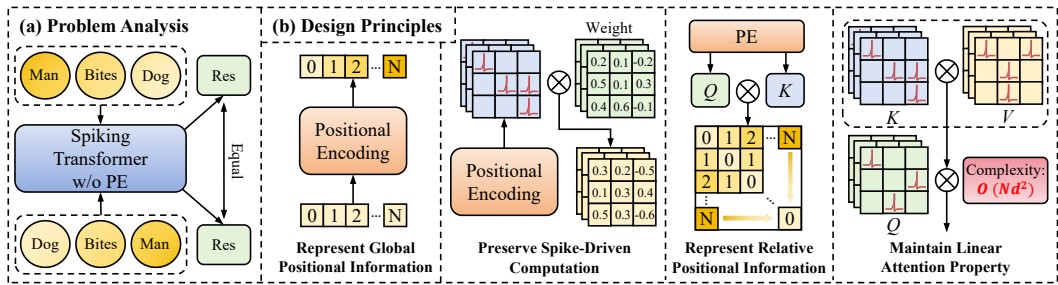

Figure 1: Problem analysis and design principles. PE refers to positional encoding.

Positional information proves crucial for sequential tasks. For instance, the sentences "Dog bites man." and "Man bites dog." convey completely opposite meanings. However, In SSA, changing the order of input tokens affects only the sequence of output tokens, while the token vector values remain unchanged. Furthermore, the MLP module mixes features only along the channel dimension rather than across tokens, which results in invariant model outputs regardless of token reordering. This behavior is clearly unreasonable for sequence modeling tasks. These observations reveal that the Spiking Transformer model has fundamental limitations in encoding positional information. Therefore, we propose introducing positional encoding mechanisms to overcome the aforementioned shortcomings.

As introduced in subsection 2.2, there has been many works on positional encoding in the ANN field. However, most of these approaches cannot be directly applied to SNN models, as incorporating them into SNNs would compromise the spike-driven characteristics of SNNs. Absolute positional encoding methods, although they can represent global positional information and are simple to implement, cannot represent relative positional information between different tokens, which prevents them from capturing the time-translational invariance property in many sequential modeling problems. While

relative positional encoding can overcome the aforementioned issues, its implementation relies on attention map computation, causing SSA to lose its linear attention property when relative positional encoding is applied. In summary, we believe that an optimal positional encoding strategy for Spiking Transformers should possess the following characteristics: (1) can represent global positional information to distinguish different positions, (2) preserve the spike-driven characteristics of SNNs, (3) have the capability to represent relative positional information, and (4) maintain the linear attention property of SSA. These four design principles are illustrated in Figure 1.

## 4.2 SPIKING POSITIONAL ENCODING

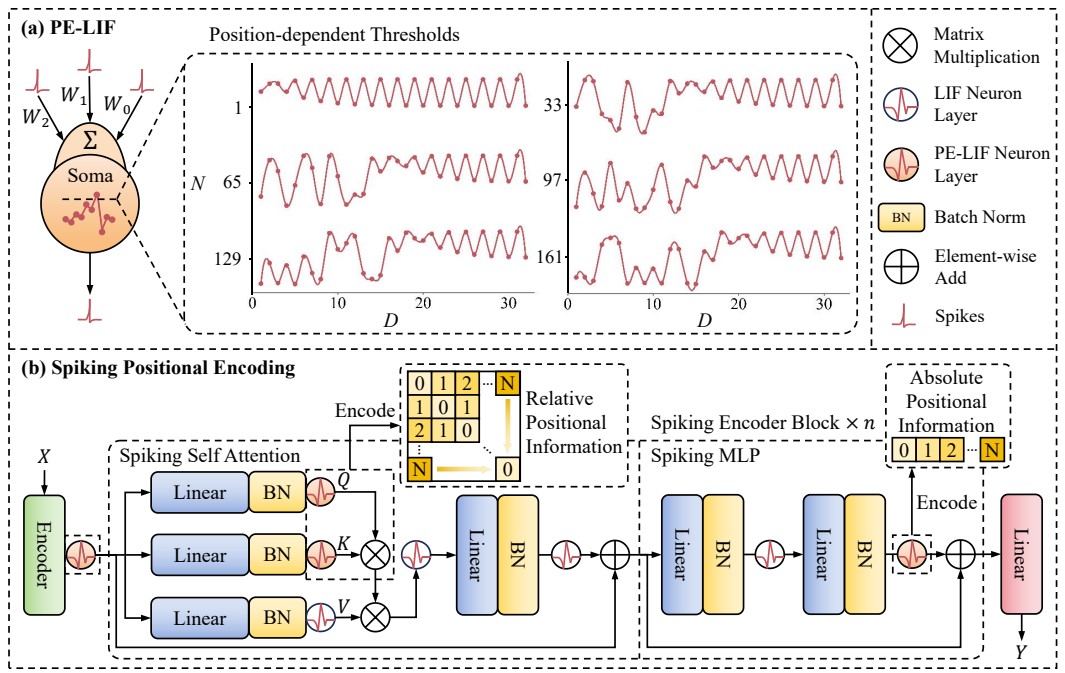

Figure 2: Overview of our method. (a) PE-LIF. We visualize the thresholds of PE-LIF neurons at some positions, demonstrating their position-dependent characteristics. (b) Spiking Positional Encoding. We demonstrate how SPE utilizes PE-LIF neuron layers to encode both absolute and relative positional information. The PE-LIF neuron layers positioned directly before SSA is responsible for encoding absolute positional information, while the PE-LIF neuron layers that activate $Q$ and $K$ are responsible for encoding relative positional information.

We design a positional encoding method that satisfies the four characteristics that an optimal positional encoding strategy for Spiking Transformers should possess as proposed in subsection 4.1, named Spiking Positional Encoding (SPE). The core of this method is the Positional Encoding Leaky Integrate-and-Fire (PE-LIF) neuron layer. The PE-LIF neuron layer is implemented based on the soft reset LIF neuron layer, with the key difference being that the PE-LIF neuron layer assigns distinct thresholds $\theta_{i,j}$ to each neuron, thereby encoding positional information into the thresholds. The positional information encoded in the thresholds is ultimately reflected in the emitted spikes through continuous spike firing and membrane potential reset processes. Specifically, the mathematical expression for $\theta_{i,j}$ in the PE-LIF neuron layer is:

$$\theta_{i,j} = \begin{cases} \theta + \lambda \cdot \cos\left(\frac{i}{10000^{\frac{j-1}{D}}}\right) & i = 1, 2, ..., N, j = 1, 3, ..., D - 1, \\ \theta + \lambda \cdot \sin\left(\frac{i}{10000^{\frac{j-2}{D}}}\right) & i = 1, 2, ..., N, j = 2, 4, ..., D. \end{cases} \tag{8}$$

where $\lambda$ is a hyperparameter used to control $\theta_{i,j}$ around $\theta$. It is worth noting that the dimension $D$ of PE-LIF neuron layer must be even, consistent with the conventional setup of Spiking Transformers.

As shown in Figure 2, in the first spike encoding layer of the entire network and the last spike encoding layers of all MLPs, we replace LIF neuron layers with PE-LIF neuron layers to encode

absolute positional information for SSA. In SSA, we replace the LIF neuron layers that activate $Q$ and $K$ with PE-LIF neuron layers to encode relative positional information. The combination of absolute positional encoding and relative positional encoding constitutes SPE. Since the thresholds of PE-LIF neurons that activate different tokens vary and are position-dependent, the PE-LIF neuron layers directly before SSA can encode global positional information into the activation values. SPE conforms to the spike-driven computing paradigm, as the spikes emitted by PE-LIF neurons are binary (0 or 1). Besides, the PE-LIF neuron layers that activate $Q$ and $K$ have the capability to encode relative positional information into the activation values, and this can be supported by the following Proposition 1. Furthermore, since PE-LIF neurons are applied separately to activate $Q$ and $K$ without requiring computation of attention maps, SPE can represent relative positional information while preserving the linear attention property of SSA.

**Proposition 1** *(Proof in Appendix A) When employing PE-LIF neuron layer to obtain activations*

$$\begin{cases} Q(t) = [\boldsymbol{q}_1(t), \boldsymbol{q}_2(t), ..., \boldsymbol{q}_N(t)]^\top \in \{0,1\}^{N \times D}, \\ K(t) = [\boldsymbol{k}_1(t), \boldsymbol{k}_2(t), ..., \boldsymbol{k}_N(t)]^\top \in \{0,1\}^{N \times D}, \end{cases} \tag{9}$$

*under the condition $\mathbb{E}\left[\hat{u}_{i,j}(t)\right] = \mathbb{E}\left[s_{i,j}(t)\right]$, there exists a function $g$ that takes $m-n$ as its argument such that $\mathbb{E}\left[\boldsymbol{q}_m^\top(t)\boldsymbol{k}_n(t)\right]$ contains $g(m-n)$, where*

$$g(m-n) = \frac{1}{2} \sum_{k=1}^{D/2} \left( \mathbb{E}\left[\mathcal{B}_{2k-1,2k-1}^{(n,m)}\right] + \mathbb{E}\left[\mathcal{B}_{2k,2k}^{(n,m)}\right] \right) \cos(m-n)\beta_k. \tag{10}$$

*Here, $\mathcal{B}^{(n,m)}$ and $\beta_k$ are defined in Appendix A. This demonstrates that PE-LIF can encode relative positional information directly into $Q(t)$ and $K(t)$.*

In addition to the four most important characteristics mentioned above, long-term decay is also an essential characteristic that positional encoding capable of representing relative positional information should possess (Su et al., 2024). Intuitively, as the distance between two tokens increases, the degree of association between them should decrease. Therefore, the value of the function representing the relative positional information between two tokens should diminish as the distance between them increases. This property is referred to as long-term decay. Proposition 2 demonstrates that the relative positional information encoded into tokens through PE-LIF possesses this desirable property.

**Proposition 2** *(Proof in Appendix B) Under certain conditions, the function $g(m-n)$ in Proposition 1 satisfies*

$$g(m-n) \sim \frac{D}{4}\mathrm{Re}\left[\int_0^1 \alpha(t)e^{i(m-n)10000^{-t}}dt\right], \tag{11}$$

*where $\alpha(t)$ is defined in Appendix B. $i$ is the imaginary unit, and $\mathrm{Re}$ denotes the real part operation. For this oscillatory integral, we have*

$$\lim_{|m-n|\to\infty} \mathrm{Re}\left[\int_0^1 \alpha(t)e^{i(m-n)10000^{-t}}dt\right] = 0, \tag{12}$$

*i.e., the function $g(m-n)$ exhibits the property of long-term decay.*

### 4.3 Membrane Potential Regularization Loss

The condition $\mathbb{E}\left[\hat{u}_{i,j}(t)\right] = \mathbb{E}\left[s_{i,j}(t)\right]$ serves as a prerequisite for Proposition 1 to hold. In general, this condition is not satisfied. To ensure its validity and thereby enable SPE to successfully encode relative positional information, we propose the Membrane Potential Regularization Loss (MPR-Loss). During training, MPR-Loss constrains the mini-batch mean of each neuron's membrane potential $\mathbb{E}_\mathcal{B}\left[\hat{u}_{i,j}(t)\right]$ to approximate the mini-batch mean of the corresponding neuron's spike output $\mathbb{E}_\mathcal{B}\left[s_{i,j}(t)\right]$, thereby ensuring that the condition $\mathbb{E}\left[\hat{u}_{i,j}(t)\right] = \mathbb{E}\left[s_{i,j}(t)\right]$ is approximately satisfied. The MPR-Loss can be formulated as:

$$\mathcal{L}_{MPR} = \frac{1}{L \times T \times N \times D} \sum_{l=1}^{L} \sum_{t=1}^{T} \sum_{i=1}^{N} \sum_{j=1}^{D} \left( \mathbb{E}_\mathcal{B}\left[\hat{u}_{i,j}^{(l)}(t)\right] - \mathbb{E}_\mathcal{B}\left[s_{i,j}^{(l)}(t)\right] \right)^2, \tag{13}$$

where $L$ denotes the number of the PE-LIF neuron layers and $T$ denotes the time steps of the PE-LIF neuron. Then the overall loss $\mathcal{L}_{Total}$ can be calculated as follows:

$$\mathcal{L}_{Total} = \begin{cases} \mathcal{L}_{CE} + \epsilon\mathcal{L}_{MPR} & \text{for classification,} \\ \mathcal{L}_{MSE} + \epsilon\mathcal{L}_{MPR} & \text{for regression,} \end{cases} \quad (14)$$

where $\mathcal{L}_{CE}$ denotes the cross-entropy loss and $\mathcal{L}_{MSE}$ represents the mean squared error loss. The coefficient $\epsilon$ balances the primary task loss (classification or regression) and the MPR-Loss. It is worth noting that MPR-Loss only acts upon the PE-LIF neuron layers that activate $Q$ and $K$.

Table 1: Experimental results of time-series forecasting on 3 benchmarks with various prediction lengths 6,24,48,96. "PE" stands for positional encoding. "Sin-PE" stands for "Sinusoidal-PE" (Vaswani et al., 2017). "Conv-PE" is the original PE in (Zhou et al., 2023b). "A" denotes absolute PE, while "R" denotes relative PE. The best results for each series are in bold font. $\uparrow$ ($\downarrow$) indicates that the higher (lower) the better.

| Dataset | Method | Spike | Type | Param (M) | Metric | Prediction Length | | | | Avg. |
|---|---|---|---|---|---|---|---|---|---|---|
| | | | | | | 6 | 24 | 48 | 96 | |
| Metr-la ($L=12$) | – | ✓ | – | 1.67 | $R^2 \uparrow$ | .697 | .491 | .383 | .242 | .453 |
| | | | | | RSE↓ | .581 | .753 | .828 | .917 | .770 |
| | Conv-PE | ✓ | A | 1.64 | $R^2 \uparrow$ | .713 | .527 | .399 | .267 | .477 |
| | | | | | RSE↓ | .565 | .725 | .818 | .903 | .753 |
| | CPG-PE | ✓ | A | 1.67 | $R^2 \uparrow$ | **.726** | .526 | .419 | .287 | .490 |
| | | | | | RSE↓ | **.553** | .720 | .806 | .890 | .742 |
| | CPG-Full | ✓ | A | 1.95 | $R^2 \uparrow$ | .719 | .530 | .417 | .286 | .488 |
| | | | | | RSE↓ | .560 | .719 | .807 | .893 | .745 |
| | **SPE** | ✓ | A&R | 1.67 | $R^2 \uparrow$ | .709 | **.543** | **.437** | **.311** | **.500** |
| | | | | | RSE↓ | .569 | **.713** | **.791** | **.875** | **.737** |
| Electricity ($L=168$) | – | ✓ | – | 1.67 | $R^2 \uparrow$ | .956 | .955 | .953 | .943 | .952 |
| | | | | | RSE↓ | .371 | .375 | .386 | .450 | .396 |
| | Sin-PE | ✗ | A | 1.67 | $R^2 \uparrow$ | .970 | .967 | .960 | .957 | .964 |
| | | | | | RSE↓ | .307 | .322 | .356 | .362 | .337 |
| | RoPE | ✗ | R | 1.67 | $R^2 \uparrow$ | .954 | .951 | .949 | .940 | .949 |
| | | | | | RSE↓ | .375 | .383 | .384 | .454 | .399 |
| | Conv-PE | ✓ | A | 1.64 | $R^2 \uparrow$ | .959 | .955 | .955 | .954 | .956 |
| | | | | | RSE↓ | .373 | .371 | .379 | .382 | .376 |
| | CPG-PE | ✓ | A | 1.67 | $R^2 \uparrow$ | .972 | .970 | .966 | .960 | .967 |
| | | | | | RSE↓ | .299 | .310 | .314 | .355 | .320 |
| | CPG-Full | ✓ | A | 1.95 | $R^2 \uparrow$ | .971 | .971 | .968 | .962 | .968 |
| | | | | | RSE↓ | .304 | .308 | .311 | .439 | .341 |
| | **SPE** | ✓ | A&R | 1.67 | $R^2 \uparrow$ | **.983** | **.972** | **.972** | **.964** | **.973** |
| | | | | | RSE↓ | **.233** | **.297** | **.297** | **.337** | **.291** |
| Solar ($L=168$) | – | ✓ | – | 1.67 | $R^2 \uparrow$ | .903 | .819 | .715 | .656 | .773 |
| | | | | | RSE↓ | .319 | .439 | .548 | .602 | .477 |
| | Sin-PE | ✗ | A | 1.67 | $R^2 \uparrow$ | .934 | .834 | .752 | .699 | .805 |
| | | | | | RSE↓ | .264 | .418 | .512 | .563 | .439 |
| | RoPE | ✗ | R | 1.67 | $R^2 \uparrow$ | .911 | .820 | .714 | .644 | .772 |
| | | | | | RSE↓ | .294 | .441 | .550 | .633 | .480 |
| | Conv-PE | ✓ | A | 1.64 | $R^2 \uparrow$ | .929 | .828 | .744 | .674 | .794 |
| | | | | | RSE↓ | .272 | .426 | .519 | .586 | .451 |
| | CPG-PE | ✓ | A | 1.67 | $R^2 \uparrow$ | **.937** | .833 | .757 | .707 | .809 |
| | | | | | RSE↓ | **.257** | .420 | .506 | .555 | .435 |
| | CPG-Full | ✓ | A | 1.95 | $R^2 \uparrow$ | .936 | .835 | .757 | .709 | .809 |
| | | | | | RSE↓ | .260 | .417 | .508 | .548 | .433 |
| | **SPE** | ✓ | A&R | 1.67 | $R^2 \uparrow$ | **.937** | **.856** | **.776** | **.721** | **.823** |
| | | | | | RSE↓ | **.257** | **.389** | **.486** | **.542** | **.419** |

## 5 EXPERIMENTS

### 5.1 DATASETS

To evaluate the effectiveness of our proposed SPE relative to other positional encoding methods, we conducted experiments on two tasks: time-series forecasting and text classification. Following (Lv et al., 2024b;a), we selected three real-world datasets for time-series forecasting: Metr-LA (Li et al., 2017), Electricity (Lai et al., 2018), and Solar (Lai et al., 2018). For text classification, following (Lv et al., 2024a; Li et al., 2021), we chose four datasets, where Subj and Waimai represent two short-text tasks, and AGNEWS (Zhang et al., 2015) and IMDB (Maas et al., 2011) represent two long-text tasks. We provide detailed information on datasets, metrics, and hyperparameter settings in Appendix C.

### 5.2 TIME-SERIES FORECASTING

The results presented in Table 1 demonstrate the comparative performance of various positional encoding methods on three time-series forecasting benchmarks across multiple prediction horizons.

The proposed SPE method consistently achieves superior performance across all three datasets, demonstrating its effectiveness for time-series forecasting tasks. SPE obtains the highest average $R^2$ scores of 0.500, 0.973, and 0.823 on Metr-la, Electricity, and Solar datasets, respectively, while simultaneously achieving the lowest average RSE values of 0.737, 0.291, and 0.419. SPE's hybrid approach that combines both absolute and relative positional information consistently outperforms other encoding methods such as Sin-PE, Conv-PE, and CPG variants, highlighting the advantage of SPE's integrated absolute-relative design.

When compared to the baseline Spikformer model without positional encoding, SPE shows substantial improvements. On the Metr-la dataset, SPE improves the average $R^2$ from 0.453 to 0.500 and reduces RSE from 0.770 to 0.737. Similar improvements are observed across all datasets, indicating the critical importance of appropriate positional encoding in spiking neural networks for temporal modeling. SPE demonstrates particularly strong performance on longer prediction horizons. This trend is consistent across all datasets, suggesting that SPE's hybrid absolute and relative encoding approach is particularly beneficial for capturing long-range temporal dependencies.

### 5.3 TEXT CLASSIFICATION

The experimental results on text classification benchmarks, as presented in the table 2, demonstrate the effectiveness of SPE across diverse sequence lengths and classification scenarios. The evaluation encompasses four datasets with varying sequence lengths: Waimai and Subj with short sequences of length 32 and 128, respectively, and longer sequences represented by AGNEWS (length 1024) and IMDB (length 2048).

SPE exhibits consistent improvements across all sequence lengths, with performance gains becoming increasingly pronounced as sequence length increases. On short sequence datasets, SPE slightly outperforms other methods. While on long sequence datasets, significant improvements are observed, where SPE achieves substantial accuracy gains of 5.34% on AGNEWS (90.04%) and 3.18% on IMDB (82.65%) compared to CPG-PE. This pattern demonstrates SPE's superior capability in modeling long-

Table 2: Accuracy on 4 text classification benchmarks.

| Dataset | Method | Spike | Acc.(%) |
|---|---|---|---|
| Waimai ($L = 32$) | – | ✓ | 86.87 |
| | Sin-PE | ✗ | 88.34 |
| | CPG-PE | ✓ | **88.49** |
| | **SPE** | ✓ | **88.49** |
| Subj ($L = 128$) | – | ✓ | 91.60 |
| | Sin-PE | ✗ | 92.20 |
| | CPG-PE | ✓ | 92.50 |
| | **SPE** | ✓ | **93.50** |
| AGNEWS ($L = 1024$) | Conv-PE | ✓ | 83.84 |
| | CPG-PE | ✓ | 84.70 |
| | **SPE** | ✓ | **90.04** |
| IMDB ($L = 2048$) | Conv-PE | ✓ | 79.08 |
| | CPG-PE | ✓ | 79.47 |
| | **SPE** | ✓ | **82.65** |

range dependencies, which is crucial for understanding complex textual patterns in extended documents, while traditional positional encoding methods show diminishing effectiveness as sequence length increases.

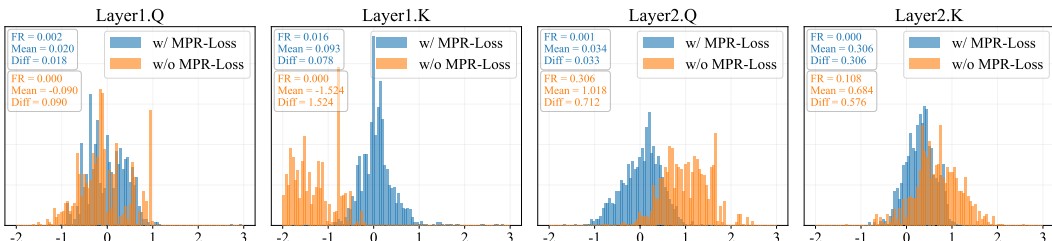

Figure 3: Comparison of membrane potential distributions in PE-LIF neurons with and without MPR-Loss. FR denotes firing rate. Mean refers to the average membrane potential. Diff represents the absolute difference between FR and Mean.

## 5.4 ABLATION STUDY

**Effectiveness of the two components of SPE.** As shown in Table 3, we conducted ablation experiments on the Solar dataset to demonstrate the effectiveness of both the absolute positional encoding (APE) and relative positional encoding (RPE) components within SPE. APE refers to the PE-LIF neuron layers positioned directly before the SSA module, while RPE refers to the PE-LIF neuron layers that activate $Q$ and $K$. Baseline is Spikformer without positional encoding. Both APE and RPE consistently outperform the baseline across different prediction lengths. And the combination of APE and RPE yields superior performance compared to using either component alone for both prediction lengths. This consistent improvement pattern indicates that absolute and relative positional information provide complementary benefits.

Table 3: Ablation study on absolute (APE) and relative (RPE) positional encoding components of SPE.

| Prediction Length | Method | Metric | |
|---|---|---|---|
| | | **$R^2$ ↑** | **RSE↓** |
| $L = 24$ | baseline | .819 | .439 |
| | w/ APE | .836 | .415 |
| | w/ RPE | .850 | .397 |
| | w/ APE & RPE | .856 | .389 |
| $L = 48$ | baseline | .715 | .548 |
| | w/ APE | .766 | .496 |
| | w/ RPE | .753 | .511 |
| | w/ APE & RPE | .776 | .486 |

**Impact of MPR-Loss.** As shown in Figure 3, after applying MPR-Loss, the expectation of membrane potential $\mathbb{E}\left[\hat{u}_{i,j}(t)\right]$ for the PE-LIF neuron layers that activate $Q$ and $K$ is closer to $\mathbb{E}\left[s_{i,j}(t)\right]$ compared to the case without MPR-Loss. Furthermore, as shown in Figure 4, an ablation study on the Solar dataset with $L = 24$ demonstrates that MPR-Loss enhances performance. Compared to the case without MPR-Loss, which achieves $R^2 = 0.828$ and RSE = 0.426, incorporating MPR-Loss improves $R^2$ by 0.022 and reduces RSE by 0.029. To avoid the influence of APE, we only used RPE in this ablation experiment. These two findings collectively validate the effectiveness of MPR-Loss.

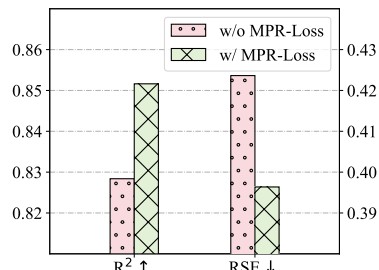

Figure 4: Ablation study on MPR-Loss

## 6 CONCLUSION

We investigate the necessity of positional encoding in Spiking Transformers and propose SPE, a method that integrates both absolute and relative positional information via the PE-LIF neuron layer. Guided by four design principles, SPE is shown theoretically to capture relative positions with desirable decay properties and is empirically validated to achieve state-of-the-art results on diverse benchmarks. This work establishes a principled and effective approach for enhancing positional representation in spiking Transformer architectures. Looking ahead, the proposed approach paves the way for building brain-inspired large language models based on Spiking Transformer architectures, offering promising opportunities for future neuromorphic intelligence.

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

## A  PROOF OF PROPOSITION 1

Let

$$
\begin{cases}
\boldsymbol{q}_m(t) = \left[ s_{m,1}^{(q)}(t), s_{m,2}^{(q)}(t), ..., s_{m,D}^{(q)}(t) \right]^T, \\
\boldsymbol{k}_n(t) = \left[ s_{n,1}^{(k)}(t), s_{n,2}^{(k)}(t), ..., s_{n,D}^{(k)}(t) \right]^T.
\end{cases}
\tag{15}
$$

Without loss of generality, we assume that all $s_{i,j}^{(q)}(t)$ and all $s_{i,j}^{(k)}(t)$ are mutually independent. Then we have

$$
\mathbb{E}\left[ \boldsymbol{q}_m^T(t)\boldsymbol{k}_n(t) \right] = \mathbb{E}\left[ \sum_{k=1}^{D} s_{m,k}^{(q)}(t) \cdot s_{n,k}^{(k)}(t) \right] = \sum_{k=1}^{D} \mathbb{E}\left[ s_{m,k}^{(q)}(t) \right] \mathbb{E}\left[ s_{n,k}^{(k)}(t) \right].
\tag{16}
$$

Since $\mathbb{E}\left[ \hat{u}_{i,j}(t) \right] = \mathbb{E}[s_{i,j}(t)]$, we have

$$
\mathbb{E}\left[ \boldsymbol{q}_m^T(t)\boldsymbol{k}_n(t) \right] = \sum_{k=1}^{D} \mathbb{E}\left[ s_{m,k}^{(q)}(t) \right] \mathbb{E}\left[ s_{n,k}^{(k)}(t) \right] = \sum_{k=1}^{D} \mathbb{E}\left[ \hat{u}_{m,k}^{(q)}(t) \right] \mathbb{E}\left[ \hat{u}_{n,k}^{(k)}(t) \right].
\tag{17}
$$

Expanding Equations 1, 2 and 4, we obtain

$$
\hat{u}_{i,j}(t) = \sum_{k=0}^{t-1} \tau^{t-1-k} I_{i,j}(k) - \theta_{i,j} \sum_{k=1}^{t-1} \tau^{t-k} s_{i,j}(k), i = 1, 2, ..., N, j = 1, 2, ..., D.
\tag{18}
$$

Furthermore, we have

$$
\begin{aligned}
\mathbb{E}\left[ \boldsymbol{q}_m^T(t)\boldsymbol{k}_n(t) \right] =& \mathbb{E}\left[ \boldsymbol{a}_m^{(q)T}(t)\boldsymbol{a}_n^{(k)}(t) \right] - \theta\mathbb{E}\left[ \boldsymbol{a}_m^{(q)T}(t)\boldsymbol{b}_n^{(k)}(t) \right] - \theta\mathbb{E}\left[ \boldsymbol{a}_n^{(k)T}(t)\boldsymbol{b}_m^{(q)}(t) \right] \\
&- \lambda\mathbb{E}\left[ \boldsymbol{a}_m^{(q)T}(t)\left( \boldsymbol{b}_n^{(k)}(t) \odot \boldsymbol{\alpha}_n \right) \right] - \lambda\mathbb{E}\left[ \boldsymbol{a}_n^{(k)T}(t)\left( \boldsymbol{b}_m^{(q)}(t) \odot \boldsymbol{\alpha}_m \right) \right] \\
&+ \theta^2\mathbb{E}\left[ \boldsymbol{b}_m^{(q)T}(t)\boldsymbol{b}_n^{(k)}(t) \right] + \theta\lambda\mathbb{E}\left[ \boldsymbol{b}_m^{(q)T}(t)\left( \boldsymbol{b}_n^{(k)}(t) \odot \boldsymbol{\alpha}_n \right) \right] \\
&+ \theta\lambda\mathbb{E}\left[ \boldsymbol{b}_n^{(k)T}(t)\left( \boldsymbol{b}_m^{(q)}(t) \odot \boldsymbol{\alpha}_m \right) \right] + \lambda^2\mathbb{E}\left[ \boldsymbol{\alpha}_m^T \mathcal{B}^{(m,n)} \boldsymbol{\alpha}_n \right],
\end{aligned}
\tag{19}
$$

where

$$
\begin{cases}
\boldsymbol{a}_i^{(\cdot)}(t) = \left[ \sum_{k=0}^{t-1} \tau^{t-1-k} I_{i,1}^{(\cdot)}(k), ..., \sum_{k=0}^{t-1} \tau^{t-1-k} I_{i,D}^{(\cdot)}(k) \right]^T, \\
\boldsymbol{b}_i^{(\cdot)}(t) = \left[ \sum_{k=1}^{t-1} \tau^{t-k} s_{i,1}^{(\cdot)}(k), ..., \sum_{k=1}^{t-1} \tau^{t-k} s_{i,D}^{(\cdot)}(k) \right]^T, \\
\boldsymbol{\alpha}_i = \left[ ..., \cos\left( \frac{i}{10000^{\frac{j-1}{D}}} \right), \sin\left( \frac{i}{10000^{\frac{j-2}{D}}} \right), ... \right]^T, \\
\mathcal{B}^{(i,j)} = \mathrm{diag}\left( \boldsymbol{b}_i^{(q)} \odot \boldsymbol{b}_j^{(k)} \right).
\end{cases}
\tag{20}
$$

Consider the term $\mathbb{E}\left[\boldsymbol{\alpha}_n^T \mathcal{B}^{(n,m)} \boldsymbol{\alpha}_m\right]$. Applying the product-to-sum formula, we obtain:

$$
\begin{aligned}
\mathbb{E}\left[\boldsymbol{\alpha}_n^T \mathcal{B}^{(n,m)} \boldsymbol{\alpha}_m\right] =& \sum_{k=1}^{D/2} \left(\mathbb{E}\left[\mathcal{B}_{2k-1,2k-1}^{(n,m)}\right] \cos m\beta_k \cos n\beta_k + \mathbb{E}\left[\mathcal{B}_{2k,2k}^{(n,m)}\right] \sin m\beta_k \sin n\beta_k\right) \\
=& \frac{1}{2} \sum_{k=1}^{D/2} \left(\mathbb{E}\left[\mathcal{B}_{2k-1,2k-1}^{(n,m)}\right] + \mathbb{E}\left[\mathcal{B}_{2k,2k}^{(n,m)}\right]\right) \cos(m-n)\beta_k \\
&+ \frac{1}{2} \sum_{k=1}^{D/2} \left(\mathbb{E}\left[\mathcal{B}_{2k-1,2k-1}^{(n,m)}\right] - \mathbb{E}\left[\mathcal{B}_{2k,2k}^{(n,m)}\right]\right) \cos(m+n)\beta_k,
\end{aligned}
\tag{21}
$$

where $\beta_i = 1/10000^{\frac{2(i-1)}{D}}$. Thus, we find that $g(m-n)$ is:

$$
g(m-n) = \frac{1}{2} \sum_{k=1}^{D/2} \left(\mathbb{E}\left[\mathcal{B}_{2k-1,2k-1}^{(n,m)}\right] + \mathbb{E}\left[\mathcal{B}_{2k,2k}^{(n,m)}\right]\right) \cos(m-n)\beta_k.
\tag{22}
$$

Therefore, Proposition 1 is proved.

## B  PROOF OF PROPOSITION 2

From Proposition 1, we have established:

$$
g(m-n) = \frac{1}{2} \sum_{k=1}^{D/2} \left(\mathbb{E}\left[\mathcal{B}_{2k-1,2k-1}^{(n,m)}\right] + \mathbb{E}\left[\mathcal{B}_{2k,2k}^{(n,m)}\right]\right) \cos(m-n)\beta_k,
\tag{23}
$$

where $\beta_i = 1/10000^{\frac{2(i-1)}{D}}$. By Euler's formula $e^{ix} = \cos x + i \sin x$, we have:

$$
g(m-n) = \frac{1}{2} \sum_{k=1}^{D/2} \left(\mathbb{E}\left[\mathcal{B}_{2k-1,2k-1}^{(n,m)}\right] + \mathbb{E}\left[\mathcal{B}_{2k,2k}^{(n,m)}\right]\right) \operatorname{Re}\left[e^{i(m-n)\beta_k}\right].
\tag{24}
$$

Utilizing the linearity of the real part of complex numbers, we obtain:

$$
\begin{aligned}
g(m-n) =& \frac{1}{2} \operatorname{Re}\left[\sum_{k=1}^{D/2} \left(\mathbb{E}\left[\mathcal{B}_{2k-1,2k-1}^{(n,m)}\right] + \mathbb{E}\left[\mathcal{B}_{2k,2k}^{(n,m)}\right]\right) e^{i(m-n)\beta_k}\right] \\
=& \frac{D}{4} \operatorname{Re}\left[\sum_{k=1}^{D/2} \left(\mathbb{E}\left[\mathcal{B}_{2k-1,2k-1}^{(n,m)}\right] + \mathbb{E}\left[\mathcal{B}_{2k,2k}^{(n,m)}\right]\right) e^{i(m-n)\beta_k} \frac{2}{D}\right].
\end{aligned}
\tag{25}
$$

From equation 25, we observe that $g(m-n)$ is precisely a Riemann sum over the interval $[0,1]$. When $D$ is large and $\alpha$ is sufficiently regular, by the definition of integration, $g(m-n)$ approximates an oscillatory integral, yielding the approximation:

$$
g(m-n) \sim \frac{D}{4} \operatorname{Re}\left[\int_0^1 \alpha(t) e^{i(m-n)10000^{-t}} dt\right],
\tag{26}
$$

where $t = \frac{2(k-1)}{D}$ and

$$
\alpha(t) = \lim_{D\to\infty} \left(\mathbb{E}\left[\mathcal{B}_{2k-1,2k-1}^{(n,m)}\right] + \mathbb{E}\left[\mathcal{B}_{2k,2k}^{(n,m)}\right]\right)\Big|_{k=\frac{Dt}{2}+1}.
\tag{27}
$$

**Lemma 1** *(Riemann-Lebesgue). Let $f(x) \in L^1(\mathbb{R})$, meaning $f$ is Lebesgue integrable on $\mathbb{R}$. Define its Fourier transform as*

$$
\hat{f}(\xi) = \int_{\mathbb{R}} f(x) e^{-i\xi x} dx,
\tag{28}
$$

*Then $\lim_{|\xi|\to\infty} \hat{f}(\xi) = 0$.*

Consider the integral

$$I(\Delta) = \int_0^1 \alpha(t)\, e^{-i\Delta 10000^{-t}}\, dt, \qquad \Delta = n - m. \tag{29}$$

Making the substitution $u = 10000^{-t}$, we have $t = -\log_{10000} u$ and $dt = -\dfrac{1}{u \ln 10000}\, du$. As $t$ ranges from 0 to 1, $u$ decreases from 1 to $10000^{-1}$. Thus,

$$I(\Delta) = \int_{u=1}^{10000^{-1}} \alpha\big(-\log_{10000} u\big)\, e^{-i\Delta u}\left(-\frac{1}{u \ln 10000}\right) du = \int_{10000^{-1}}^{1} \beta(u)\, e^{-i\Delta u}\, du, \tag{30}$$

where we define

$$\beta(u) = \frac{\alpha\big(-\log_{10000} u\big)}{u \ln 10000}. \tag{31}$$

Assuming $\alpha(t)$ is bounded on $[0, 1]$, we have $\beta(u) \in L^1([10000^{-1}, 1])$. Since $1/(u \ln 10000)$ is integrable on $[10000^{-1}, 1]$, $\beta$ is integrable. Therefore, by Lemma 1, we obtain

$$\lim_{|\Delta| \to \infty} I(\Delta) = 0 \quad \implies \quad \lim_{|\Delta| \to \infty} \operatorname{Re} I(\Delta) = 0. \tag{32}$$

Therefore, Proposition 2 is proved.

## C  EXPERIMENTAL SETTINGS

### C.1  TIME-SERIES FORECASTING

In the time-series forecasting experiments, all positional encoding methods employ a two-layer Spikformer architecture with 256 dimensions as their base model. The simulation time step for neurons is set to 4. The learning rate is set to 0.001, batch size to 64, and the AdamW optimizer is utilized. An early stopping strategy with a tolerance of 50 epochs is adopted. Weight decay is configured with different values according to the degree of overfitting observed in each dataset. $\lambda$ is set to 0.3 and $\epsilon$ to $10^{-4}$. The random seed is fixed at 42. For the METR-LA dataset, the observation length is set to 12 for short-term forecasting, while for the Solar Energy and Electricity datasets, the observation length is set to 168 for long-term forecasting. All datasets are evaluated across four different prediction lengths: 6, 24, 48, and 96. The parameter results reported in Table 1 are measured under the prediction length of 6. The details of each dataset are provided as follows.

**METR-LA Dataset**  The METR-LA (Metropolitan Traffic-Los Angeles) dataset represents a comprehensive collection of traffic flow measurements from the Los Angeles Metropolitan transportation network. This short-term time-series dataset comprises 34,272 samples across 207 variables. The dataset captures traffic sensor readings from various locations throughout the Los Angeles freeway system, providing granular insights into urban traffic patterns. The variables typically include traffic flow rates, occupancy percentages, and speed measurements from different sensor stations. With a train-validation-test split ratio of 0.7:0.2:0.1, this dataset serves as a benchmark for evaluating traffic forecasting models and understanding spatiotemporal dependencies in urban transportation networks.

**Solar Energy Dataset**  The Solar Energy dataset constitutes a long-term meteorological time-series collection focused on solar irradiance measurements across multiple geographical locations. This dataset comprises 52,560 samples with 137 variables representing measurements of global horizontal irradiance, direct normal irradiance, and diffuse horizontal irradiance from solar monitoring stations distributed across different climatic zones. The variables may also incorporate auxiliary meteorological parameters such as temperature, humidity, and cloud cover indices that influence solar energy generation. With a 0.6:0.2:0.2 split for training, validation, and testing, this dataset provides a robust foundation for developing solar energy forecasting models and understanding the complex temporal dynamics of renewable energy resources.

**Electricity Dataset**   The Electricity dataset represents a comprehensive time-series collection of electrical power consumption measurements from multiple consumers or regions. This long-term dataset contains 26,304 samples across 321 variables capturing hourly electricity usage data from residential, commercial, and industrial consumers, providing insights into diverse consumption behaviors and load profiles. The variables often include power consumption measurements from different customer segments, geographic regions, or utility substations, allowing for multi-dimensional analysis of electrical demand patterns. Employing a 0.6:0.2:0.2 distribution for training, validation, and testing phases, this dataset serves as a valuable resource for developing electricity demand forecasting models and understanding the complex temporal dependencies in power system operations.

In this experiment, we employ the coefficient of determination ($R^2$) and the Root Relative Squared Error (RSE) as evaluation metrics. $R^2$ is a statistical measure that quantifies the proportion of variance in the dependent variable that is predictable from the independent variables in a regression model. In the context of time-series forecasting, $R^2$ evaluates how well the predicted values explain the variability observed in the actual time-series data. Mathematically, $R^2$ is defined as:

$$R^2 = 1 - \frac{SS_{res}}{SS_{tot}}, \tag{33}$$

where $SS_{res} = \sum_{t=1}^{n}(y_t - \hat{y}_t)^2$ represents the residual sum of squares. $SS_{tot} = \sum_{t=1}^{n}(y_t - \bar{y})^2$ represents the total sum of squares. $y_t$ denotes the actual value at time $t$. $\hat{y}_t$ denotes the predicted value at time $t$. $\bar{y}$ represents the mean of actual values. $n$ is the number of observations. $R^2$ ranges from 0 to 1, where values closer to 1 indicate superior predictive performance. However, $R^2$ can be negative when the model performs worse than a naive baseline that simply predicts the mean value for all observations. RSE is a normalized error metric that measures the relative magnitude of prediction errors in time-series forecasting. RSE provides a dimensionless measure of forecast accuracy by normalizing the root mean squared error against a reference baseline. The RSE is formally expressed as:

$$\text{RSE} = \sqrt{\frac{\sum_{t=1}^{n}(y_t - \hat{y}_t)^2}{\sum_{t=1}^{n}(y_t - \bar{y})^2}}, \tag{34}$$

where the notation follows the same conventions as defined above. Alternatively, RSE can be expressed in terms of the coefficient of determination:

$$\text{RSE} = \sqrt{1 - R^2} \tag{35}$$

RSE values range from 0 to infinity, where smaller values indicate better forecasting performance. An RSE value of 0 represents perfect prediction, while values greater than 1 suggest that the model performs worse than the naive baseline of predicting the historical mean. The metric is particularly valuable for comparing forecast accuracy across different time series with varying scales and magnitudes.

C.2   TEXT CLASSIFICATION

In the text classification experiments, we employed a 6-layer SpikeBERT model with a hidden dimension of 768 (Lv et al., 2023). The neuronal simulation time step was set to 4. For the Waimai and Subj datasets, the learning rate was set to $5 \times 10^{-4}$, with a batch size of 32 and maximum sequence lengths of 32 and 128, respectively. For the AGNEWS dataset, we used a learning rate of $1.7 \times 10^{-4}$, a batch size of 12, and a maximum sequence length of 1024. For the IMDB dataset, the learning rate was set to $1.25 \times 10^{-4}$, the batch size to 8, and the maximum sequence length to 2048. Across all four datasets, we adopted cosine annealing for learning rate scheduling and fixed the random seed to 42. $\lambda$ is set to 0.3 and $\epsilon$ to $10^{-4}$. The weight decay was tuned individually for each dataset to mitigate overfitting. The details of each dataset are provided as follows.

**Waimai Dataset**   The Waimai dataset is a Chinese sentiment analysis corpus comprised of user reviews collected from a food delivery platform. The dataset contains approximately 11,987 review samples with a binary sentiment classification structure, including around 4,000 positive reviews and nearly 8,000 negative reviews, creating an imbalanced distribution that reflects real-world sentiment patterns.

**Subj Dataset**   The Subj dataset is a widely-used benchmark for subjectivity detection in natural language processing. This binary classification dataset comprises 10,000 sentences evenly divided between subjective and objective instances, with 5,000 subjective sentences extracted from movie review snippets on Rotten Tomatoes and 5,000 objective sentences derived from plot summaries on the Internet Movie Database.

**AGNEWS Dataset**   The AGNEWS dataset is a subset of AG's Corpus of news articles, constructed by assembling titles and description fields from the four largest classes: "World," "Sports," "Business," and "Science/Technology". The dataset contains 30,000 training samples and 1,900 test samples per class, totaling 120,000 training examples and 7,600 test examples across all categories.

**IMDB Dataset**   The IMDB Movie Review Dataset, also known as the Large Movie Review Dataset, is a comprehensive corpus designed for binary sentiment classification, containing substantially more data than previous benchmark datasets in the field. The dataset provides 25,000 highly polar movie reviews for training and 25,000 for testing, with each review labeled as either positive or negative sentiment.

## D   MORE COMPARISONS OF MEMBRANE POTENTIAL DISTRIBUTIONS

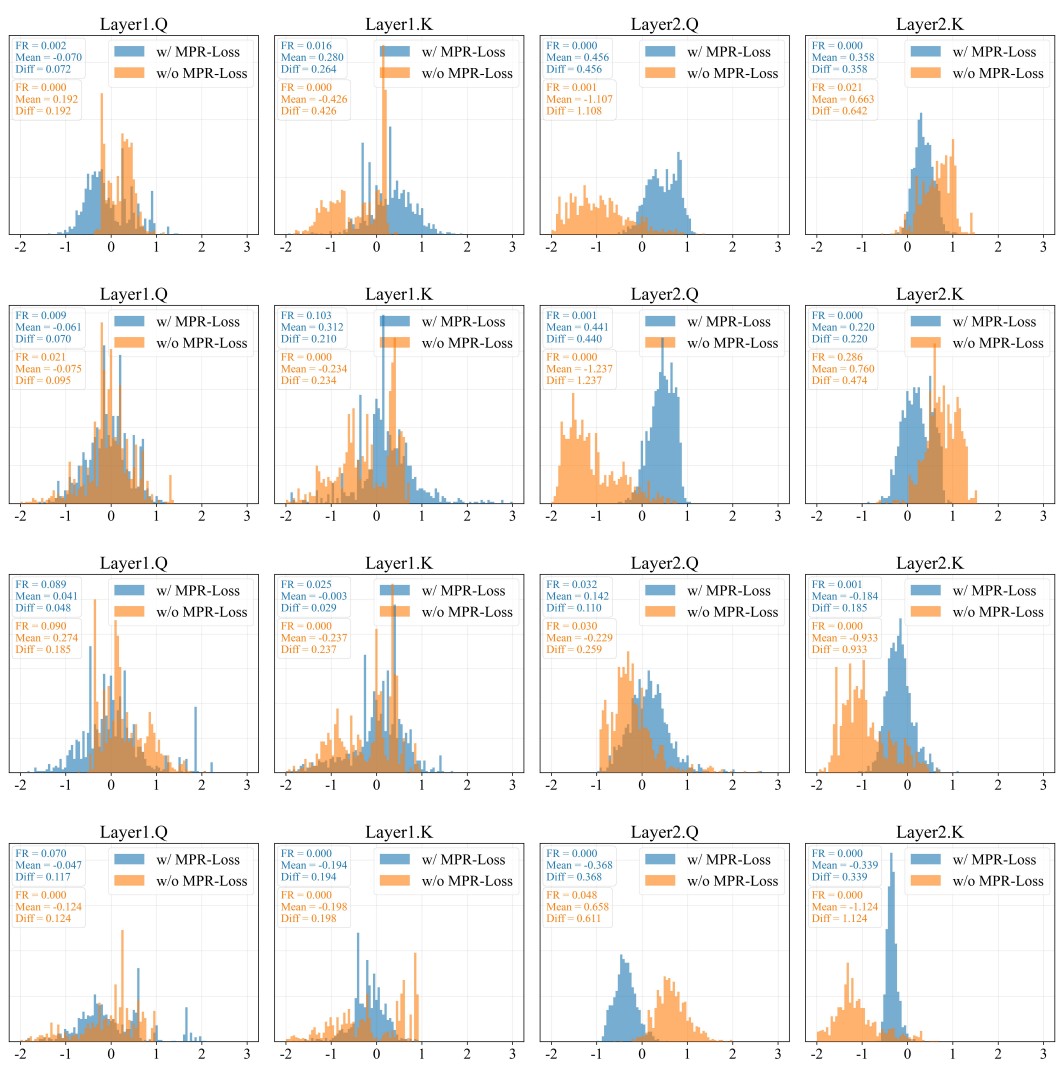

Figure 5: More comparisons of membrane potential distributions.

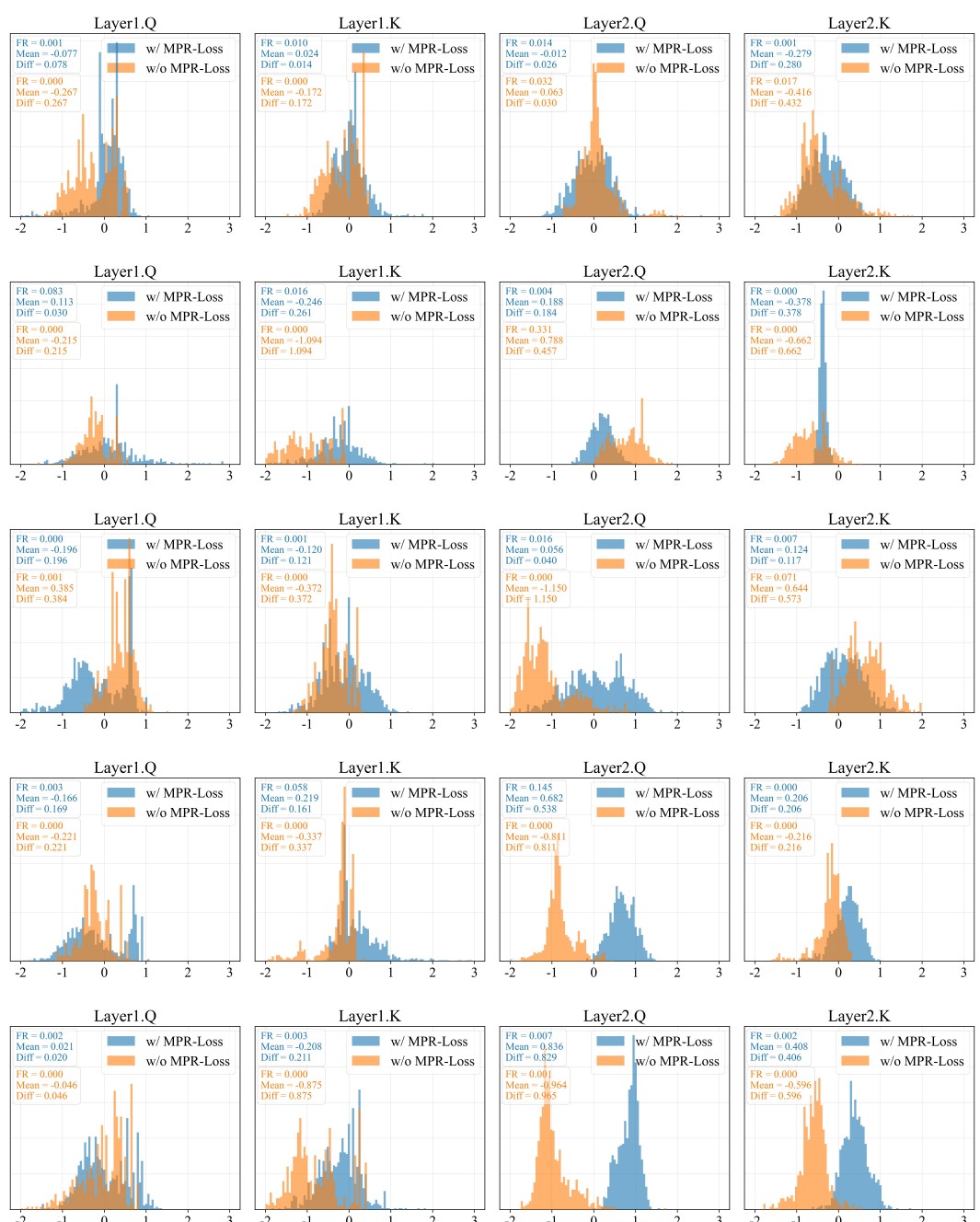

Figure 5: More comparisons of membrane potential distributions (continued).

To further validate the effectiveness of MPR-Loss in constraining the membrane potentials of PE-LIF neurons, we present additional comparisons of membrane potential distributions with and without MPR-Loss in Figure 5. The results consistently demonstrate that MPR-Loss effectively drives $\mathbb{E}\left[\hat{u}_{i,j}(t)\right]$ toward $\mathbb{E}\left[s_{i,j}(t)\right]$.

## E  THE USE OF LARGE LANGUAGE MODELS

We used large language models only to improve the writing of this manuscript. It did not contribute to the research ideas, methods or experiments. All scientific contributions are the authors' own work.