# OpenReview forum: "Positional Encoding for Spiking Transformers"
_ICLR.cc/2026/Conference — Submitted to ICLR 2026_

### Official Review · Reviewer_6QkJ · 2025-10-26

**Soundness:** 2
**Presentation:** 3
**Contribution:** 3
**Rating:** 4
**Confidence:** 4

**Summary:**

The authors address the lack of absolute and relative positional encoding in Spike Transformers and, drawing inspiration from positional encoding designs in the ANN domain, propose a set of design principles for positional encoding in SNNs. Based on these principles, they design a novel neuron model, PE-LIF, which integrates absolute positional information into its threshold dynamics. Theoretical analysis further demonstrates that this neuron model, while encoding absolute positional information, is also capable of representing relative positional information. Moreover, the authors introduce an MPR loss to maintain the theoretical assumptions. Finally, extensive experimental results are presented to validate the effectiveness of the proposed method.

**Strengths:**

1. The motivation of this paper is clear and well-defined, directly targeting a major pain point in the current Spike Transformer research — the lack of positional encoding, which in turn limits the effectiveness of SNNs in sequential modeling.

2. The paper provides thorough analysis and well-founded theoretical proofs, accompanied by comprehensive experiments that validate the proposed method’s effectiveness across multiple datasets.

**Weaknesses:**

1. The letter notations in the paper are somewhat inconsistent. In Section 3.1, t denotes the time step, N denotes the number of tokens, and D represents the dimension; in Section 4.3, L refers to the number of PE-LIF neuron layers; yet in Table 2, L is again used to represent the input length.

2. The novelty of this paper remains open to question. Although the problem it addresses is indeed an important one in the SNN field, the proposed method is largely based on RoPE — both in terms of the underlying formulation and the way absolute positional encoding is extended to relative positional encoding through the attention mechanism. Moreover, there is no ablation study specifically analyzing the effect of the newly proposed MPR loss; only a comparison of one related metric is provided, and the improvement is not particularly significant. Overall, the work appears to focus primarily on the PE-LIF neuron design derived from RoPE, rather than introducing a fundamentally new mechanism.

3. The remaining details can be found in the Questions section below.

**Questions:**

1. In Proposition 2, the authors state that as the relative distance between tokens increases, their dependency weakens and eventually approaches zero, thereby suggesting that the proposed encoding method enables dependency decay with distance. However, wouldn’t proving monotonicity or approximate monotonicity better demonstrate this characteristic?

2. In Proposition 1, the authors introduce a hypothesis, and in Section 4.2 they incorporate the MPR loss to ensure that this hypothesis is approximately satisfied during training. Could the authors visualize the variation of MPR loss throughout the training process?

3. According to Figure 4, the introduction of MPR loss seems to have little impact on the R² and RSE losses. Have the authors conducted an ablation study where MPR loss is removed and only PE-LIF is considered?

4. In this paper, the variable T used during sequence modeling with SNNs appears to be obtained through repetitive encoding. Although such repeated encoding is widely used in image processing, its role in sequential modeling is not entirely clear. Could the authors further explain the rationale behind adopting this approach for sequences?

5. The authors encode positional information into the thresholds of spiking neurons, resulting in each neuron having a distinct threshold. Since the number of neurons is directly related to the input token length, could the authors discuss the implementation complexity of such independently parameterized thresholds when deploying the model on actual neuromorphic hardware?

---

> ### Author Response · Authors · 2025-11-20
> **Responses to weakness 1 & 2**
>
> We are deeply grateful for your time in reviewing our manuscript. Your insights will be of great help to us in improving the quality of our paper.
>
> > *Weakness 1: The letter notations in the paper are somewhat inconsistent. In Section 3.1, t denotes the time step, N denotes the number of tokens, and D represents the dimension; in Section 4.3, L refers to the number of PE-LIF neuron layers; yet in Table 2, L is again used to represent the input length.*
>
> Response to weakness 1: Thank you for pointing out the inconsistency in the notation. We will revise the symbol $L$ in Table 2 to $N$.
>
> > *Weakness 2: The novelty of this paper remains open to question. Although the problem it addresses is indeed an important one in the SNN field, the proposed method is largely based on RoPE — both in terms of the underlying formulation and the way absolute positional encoding is extended to relative positional encoding through the attention mechanism. Moreover, there is no ablation study specifically analyzing the effect of the newly proposed MPR loss; only a comparison of one related metric is provided, and the improvement is not particularly significant. Overall, the work appears to focus primarily on the PE-LIF neuron design derived from RoPE, rather than introducing a fundamentally new mechanism.*
>
> Response to weakness 2: We acknowledge that our method is inspired by positional encoding approaches in ANNs, including RoPE. However, our method is **fundamentally different** from RoPE.
>
> RoPE takes the following form:
>
> $$
> q_n^{\text{rope}} = \mathcal{R}_n q_n,\quad k_n^{\text{rope}} = \mathcal{R}_n k_n,
> $$
>
> where $\mathcal{R}$ is a block-diagonal rotation matrix used to encode positional information. In contrast, our method is essentially an additive positional encoding scheme, which is entirely different from RoPE’s rotation-based encoding mechanism.
> Moreover, our method leverages intrinsic properties of spiking neurons and thus preserves the event-driven computation paradigm of SNNs, whereas RoPE and related approaches in ANNs cannot be directly applied to SNNs.
>
> Regarding *the underlying formulation*, the choice of sinusoidal functions is due to their inherent ability to represent relative positional information (see Eq. 21) as well as their natural long-range decay behavior (see Proposition 2 & Appendix B). This formulation has a long history, first introduced alongside the Transformer in [1] as sinusoidal positional encoding, and many subsequent positional encoding methods have adopted similar structures.
>
> As for *extending absolute positional encoding to relative positional encoding through the attention mechanism*, this idea is indeed inspired by RoPE, but the actual formulation differs significantly. RoPE computes the attention map as:
>
> $$
> q_n^{\text{rope}} \cdot k_m^{\text{rope}} = q_n^{T} \mathcal{R}_n^{T} \mathcal{R}_m k_m = q_n^{T} \mathcal{R} _{m-n} k_m,
> $$
>
> where RoPE uses the rotation matrix $\mathcal{R}_{m-n}$ to represent relative positional information. However, in spiking Transformer models, directly multiplying $q$ and $k$ by rotation matrices would compromise the event-driven computation characteristics.
> In our method, positional information is instead encoded into the thresholds of spiking neurons, and relative positional information arises through the function $g(m-n)$ embedded in $\mathbb{E}[q_n^T k_m]$ (see Propostion 1 & Appendix A).
>
> Additional ablation results for the MPR-Loss is provided in our response to question 3.
>
> > [1] *Ashish Vaswani, et al. Attention Is All You Need. NeurIPS 2017.*

---

> ### Author Response · Authors · 2025-11-20
> **Responses to question 1, 2 & 3**
>
> > *Question 1: In Proposition 2, the authors state that as the relative distance between tokens increases, their dependency weakens and eventually approaches zero, thereby suggesting that the proposed encoding method enables dependency decay with distance. However, wouldn’t proving monotonicity or approximate monotonicity better demonstrate this characteristic?*
>
> Response to question 1: Equation (11) is an oscillatory integral. Such integrals may contain intervals where the function is monotonically increasing and therefore **do not satisfy a globally monotonic-decreasing property**. Nevertheless, the overall trend decreases as $|m-n|$ grows.
>
> For example, assume $\alpha(t)$ is a constant and consider the function
> $$
> f(x) = \mathrm{Re}\left[\int_0^1 e^{i x \cdot 10000^{-t}} dt\right],
> $$
> where $x = m - n > 0$. Since the real part corresponds to the cosine component, we have
> $$
> f(x) = \int_0^1 \cos(x \cdot 10000^{-t}) dt.
> $$
> Let
> $$
> u = 10000^{-t} \quad \Rightarrow \quad du = -u \ln(10000)dt.
> $$
> so
> $$
> dt = -\frac{du}{u \ln(10000)}.
> $$
> When $t = 0$, $u = 1$; and when $t = 1$, $u = 10000^{-1} = \frac{1}{10000}$. Thus, the integral becomes
> $$
> f(x)
> = \int_1^{1/10000} \cos(xu)\left(-\frac{1}{u\ln(10000)}\right)du
> = \frac{1}{\ln(10000)} \int_{1/10000}^1 \frac{\cos(xu)}{u}du.
> $$
>
> Differentiating under the integral sign gives
> $$
> f'(x)
> = \frac{1}{\ln(10000)}\int_{1/10000}^1 \frac{\partial}{\partial x}\left(\frac{\cos(xu)}{u}\right)du
> = \frac{1}{\ln(10000)} \int_{1/10000}^1 -\sin(xu)du.
> $$
> Since
> $$
> \int \sin(xu)du = -\frac{\cos(xu)}{x},
> $$
> we obtain
> $$
> \int_{1/10000}^1 \sin(xu)du
> = -\frac{\cos(x)}{x} + \frac{\cos(x/10000)}{x}.
> $$
> Substituting into the derivative yields
> $$
> f'(x)
> = \frac{\cos(x) - \cos(x/10000)}{x\ln(10000)}.
> $$
> Because $x>0$ and $\ln(10000)>0$, the sign of $f'(x)$ is determined by
> $$
> \cos(x) - \cos(x/10000).
> $$
> Taking $x = 2\pi k$ $(k\in\mathbb{N})$,
> $$
> \cos(2\pi k)=1,\qquad
> \cos\left(\frac{2\pi k}{10000}\right)
> = \cos\left(\frac{\pi k}{5000}\right).
> $$
> If $\frac{\pi k}{5000}$ is close to $\pi$, then
> $$
> \cos\left(\frac{\pi k}{5000}\right)\approx -1,
> $$
> and hence
> $$
> \cos(x) - \cos(x/10000) \approx 1 - (-1) = 2 > 0,
> $$
> implying $f'(x)>0$, i.e., the function is increasing.
> **Since there exist intervals where $f'(x) > 0$, the function $f(x)$ is not monotonically decreasing for $x>0$.** Because of the existence of the above counter example, we cannot prove the monotonicity of Equation (11).
>
> Although $f(x)$ is not strictly monotonic, numerical plots generated using *Mathematica* do exhibit an approximately monotonic-decreasing behavior.
>
>
> > *Question 2: In Proposition 1, the authors introduce a hypothesis, and in Section 4.2 they incorporate the MPR loss to ensure that this hypothesis is approximately satisfied during training. Could the authors visualize the variation of MPR loss throughout the training process?*
>
> Response to question 2:
>
> |epoch|1|21|41|61|81|101|121|141|161|181|201|
> |-|-|-|-|-|-|-|-|-|-|-|-|
> |MPR-Loss|0.0572|0.0474|0.0450|0.0423|0.0445|0.0465|0.0468|0.0473|0.0469|0.0463|0.0455|
>
> Since the rebuttal does not allow image uploads, we present the trend of the MPR-Loss during training in tabular form. As shown, the MPR-Loss decreases rapidly in the early training stage and later oscillates within a certain range. These results indicate that the MPR-Loss contributes to regulating the distribution of membrane potentials.
>
> The above experiments were conducted on the Electricity dataset with a prediction length of 48.
>
>
> > *Question 3: According to Figure 4, the introduction of MPR loss seems to have little impact on the R² and RSE losses. Have the authors conducted an ablation study where MPR loss is removed and only PE-LIF is considered?*
>
> Response to question 3:
>
> |Method|R$^2_{PL=6}$|RSE$_{PL=6}$|R$^2_{PL=24}$|RSE$_{PL=24}$|R$^2_{PL=48}$|RSE$_{PL=48}$|R$^2_{PL=96}$|RSE$_{PL=96}$|
> |-|-|-|-|-|-|-|-|-|
> |w/ MPR-Loss|**0.983**|**0.233**|**0.972**|**0.297**|**0.972**|**0.297**|**0.964**|**0.337**|
> |w/o MPR-Loss|0.966|0.326|0.959|0.360|0.957|0.368|0.946|0.414|
>
> We conducted experiments on the Electricity dataset under prediction lengths of 6, 24, 48, and 96. In this study, we compared two settings: using both MPR-Loss and PE-LIF, and using only PE-LIF. **The results show that incorporating MPR-Loss consistently improves performance across all settings, demonstrating its effectiveness.**
>
> In the table above, $PL$ denotes the prediction length.

---

> ### Author Response · Authors · 2025-11-20
> **Responses to question 4 & 5**
>
> > *Question 4: In this paper, the variable T used during sequence modeling with SNNs appears to be obtained through repetitive encoding. Although such repeated encoding is widely used in image processing, its role in sequential modeling is not entirely clear. Could the authors further explain the rationale behind adopting this approach for sequences?*
>
> Response to question 4: Our understanding is that, for both image processing and sequence tasks, LIF neurons encode information through their firing rates. This form of representation is analogous to activation quantization, with the key difference that LIF neurons inherently support event-driven computation and exhibit temporal dynamics.
>
> > *Question 5: The authors encode positional information into the thresholds of spiking neurons, resulting in each neuron having a distinct threshold. Since the number of neurons is directly related to the input token length, could the authors discuss the implementation complexity of such independently parameterized thresholds when deploying the model on actual neuromorphic hardware?*
>
> Response to question 5: Current research on deploying SNNs primarily focuses on FPGAs, and FPGA development workflows are often used as an important reference for designing and validating neuromorphic chips. Therefore, we discuss this issue using FPGA deployment as an example. In an FPGA, the threshold matrix required by our method can be stored in off-chip RAM. When a processing element (PE) needs a threshold at a specific position, the value is first fetched from off-chip RAM into on-chip RAM, and then read by the PE. The transfer from off-chip RAM to on-chip RAM is serialized. Assuming an off-chip RAM clock frequency of 250 MHz and a data width of 16 bits, transmitting a $128 \times 768$ threshold matrix of 16-bit values incurs a latency of approximately 196.608 µs, which is negligible. The subsequent on-chip RAM–to-PE access can run in parallel with weight fetching (through multiple read channels), introducing no additional latency. In summary, the memory-access procedure for threshold matrices is highly similar to that for weight parameters, and **thus introduces no extra hardware complexity, resource overhead, or access latency**.
>
> If you have any remaining concerns, please feel free to continue the discussion with us. Once again, I sincerely appreciate your valuable contributions to improving the quality of this paper.

---

> ### Author Response · Authors · 2025-11-27
>
> Dear Reviewer 6QkJ,
>
> Thank you very much for your time in reviewing our manuscript and for your valuable comments. As the author–reviewer discussion phase is drawing to a close, we would like to confirm whether our responses have satisfactorily addressed your concerns. We have provided detailed point-by-point replies, and we hope they resolve the issues you raised. Should you require any further clarification or have additional questions, please do not hesitate to let us know. We would be pleased to continue the discussion.
>
> Best regards.

---

### Official Review · Reviewer_SaAm · 2025-10-29

**Soundness:** 3
**Presentation:** 2
**Contribution:** 2
**Rating:** 4
**Confidence:** 5

**Summary:**

This paper proposes a novel positional encoding approach for Spiking Transformers; however, the experimental verification is insufficient.

**Strengths:**

The paper identifies a fundamental limitation in directly applying ANN-based positional encodings to Spiking Neural Networks (SNNs), effectively motivating the need for SNN-specific solutions. and proposes an Integer-firing neuron for positional encoding in spiking transformers.

**Weaknesses:**

1. The explanation of the complexity of SSA in the paper is unsatisfactory(shown in "A key advantage of SSA lies in its linear attention property, whereby the time complexity of SSA, line 179-184). This paper focuses on "Positional Encoding for Spiking Transformers"; however, the spiking self-attention in spiking transformers (or self-attention in general) is primarily designed for large models, where the embedding dimension $ D$ is substantial, and $D^2$ is certainly not negligible.
2. The model used in the experiments is too small（1-2M） to thoroughly validate the effectiveness of the positional encoding in the Spiking Transformer. Moreover, the experiments are relatively simple — for instance, this paper lacks evaluations on more widely recognized time series benchmarks such as the ETT dataset.
3. Lack of comparison with the SOTA ANN methods.
4. Lack of discussion on deployment on neural chips.
5. Lack of comparison with "Toward Relative Positional Encoding in Spiking Transformers"

**Questions:**

see above

---

> ### Author Response · Authors · 2025-11-25
> **Responses to weakness 1, 2, 3 & 4**
>
> We are deeply grateful for your time in reviewing our manuscript. Your insights will be of great help to us in improving the quality of our paper.
>
> > *Weakness 1: The explanation of the complexity of SSA in the paper is unsatisfactory(shown in "A key advantage of SSA lies in its linear attention property, whereby the time complexity of SSA, line 179-184). This paper focuses on "Positional Encoding for Spiking Transformers"; however, the spiking self-attention in spiking transformers (or self-attention in general) is primarily designed for large models, where the embedding dimension $D$ is substantial, and $D^2$ is certainly not negligible.*
>
> Response to weakness 1: For BERT-style models, the sequence length $N$ is typically 512 and the hidden dimension $D$ is typically 768, so the condition $D \ll N$ is not satisfied. In contrast, for GPT-style models such as Qwen3-32B, the hidden dimension is $D = 4096$, while the maximum supported context length is $N = 32768$. In this case, $N$ is eight times larger than $D$. Consequently, applying linear attention to Qwen3-32B can reduce the time complexity of the attention mechanism by roughly eight times, resulting in a substantial efficiency gain. Moreover, in the era of **Large Language Models**, the demand for long-context processing is expected to continue increasing, so the efficiency advantages of linear attention are likely to become increasingly significant.
>
> > *Weakness 2: The model used in the experiments is too small（1-2M） to thoroughly validate the effectiveness of the positional encoding in the Spiking Transformer. Moreover, the experiments are relatively simple — for instance, this paper lacks evaluations on more widely recognized time series benchmarks such as the ETT dataset.*
>
> Response to weakness 2: In the NLP experiment shown in Table 2, we employed a 6-layer, 768-dimensional Spikformer model, which has a total parameter count of 68.42M, indicating that it is a relatively large model.
>
> |Method|R$^2_{h1}$|RSE$_{h1}$|R$^2_{h2}$|RSE$_{h2}$|R$^2_{m1}$|RSE$_{m1}$|R$^2_{m2}$|RSE$_{m2}$|
> |-|-|-|-|-|-|-|-|-|
> |Baseline|0.247|0.958|0.875|1.128|0.379|0.870|0.936|0.799|
> |**Ours**|**0.575**|**0.720**|**0.913**|**0.939**|**0.766**|**0.537**|**0.961**|**0.623**|
>
> We conduct experiments on the ETT datasets, and the results are presented in the table above. Our method consistently outperform the baseline across different time granularities, further demonstrating the effectiveness of our approach. The experiments are configured with a prediction length of 24 time steps and an observation length of 168 time steps.
>
> > *Weakness 3: Lack of comparison with the SOTA ANN methods.*
>
> Response to weakness 3: In Table 1, we present a comparison of the experimental results between our method and RoPE [1] (the SOTA ANN method). The experimental results demonstrate that our method outperforms RoPE when applied to Spikformer model.
>
> Although RoPE performs well in the ANN domain, its full-precision rotation matrix is not compatible with the Spiking Self-Attention (SSA) mechanism unique to SNNs. For example, in SSA, the attention map $Q^TK$ approximately follows a binomial distribution, which exhibits a heavy-tailed property that is suitable for attention scores. However, incorporating RoPE may disrupt this binomial distribution. Since there is no Softmax operation in SSA, the distribution will no longer maintain its heavy-tailed nature. In such a case, RoPE has a detrimental effect on the performance of SNN models. **Therefore, our method shows superior performance on SNNs compared to RoPE.**
>
> > [1] *Su, et al. RoFormer: Enhanced Transformer with Rotary Position Embedding. Neurocomputing.*
>
> > *Weakness 4: Lack of discussion on deployment on neural chips.*
>
> Response to weakness 4: Current research on deploying SNNs primarily focuses on FPGAs, and FPGA development workflows are often used as an important reference for designing and validating neuromorphic chips. Therefore, we discuss this issue using FPGA deployment as an example. In an FPGA, the threshold matrix required by our method can be stored in off-chip RAM. When a processing element (PE) needs a threshold at a specific position, the value is first fetched from off-chip RAM into on-chip RAM, and then read by the PE. The transfer from off-chip RAM to on-chip RAM is serialized. Assuming an off-chip RAM clock frequency of 250 MHz and a data width of 16 bits, transmitting a $128 \times 768$ threshold matrix of 16-bit values incurs a latency of approximately 196.608 µs, which is negligible. The subsequent on-chip RAM–to-PE access can run in parallel with weight fetching (through multiple read channels), introducing no additional latency. In summary, the memory-access procedure for threshold matrices is highly similar to that for weight parameters, and **thus introduces no extra hardware complexity, resource overhead, or access latency**.

---

> > ### Comment · Reviewer_SaAm · 2025-11-27
> > **$D^2$=$4096 \times 4096 $ Vs. $N = 32768$.**
> >
> > As you said,  Qwen3-32B, the hidden dimension is $D = 4096$, the maximum supported context length is $N = 32768$. $D^2$=$4096 \times 4096 $. The Complexity of SSA is $O(ND^2)$  can not be seen as linear complexity O(N). Therefore, I believe that the description of O(N) in your paper is a fatal error in expression, consistent with my previous review comments.

---

> ### Author Response · Authors · 2025-11-25
> **Response to weakness 5**
>
> > *Weakness 5: Lack of comparison with "Toward Relative Positional Encoding in Spiking Transformers"*
>
> |Method|Waimai|Subj|AGNEWS|IMDB|
> |-|-|-|-|-|
> |Gray-PE|88.40|92.50|84.92|79.79|
> |Log-PE|88.46|92.80|86.77|80.46|
> |**SPE**|**88.49**|**93.50**|**90.04**|**82.65**|
>
> Response to weakness 5: The results in the table above show a comparison between our method and the two methods from [1] across four NLP datasets.
>
> > [1] *Lv, et al. Toward Relative Positional Encoding in Spiking Transformers. NeurIPS 2025.*
>
> If you have any remaining concerns, please feel free to continue the discussion with us. Once again, I sincerely appreciate your valuable contributions to improving the quality of this paper.

---

> ### Author Response · Authors · 2025-11-27
>
> First, we would like to thank you for your response. However, we believe there may be a misunderstanding regarding our previous explanation, and we would like to clarify it further.
>
> Under the Qwen3-32B setting, when using the **non-linear** attention mechanism, the time complexity is
>
> $$
> N^2D = 32768 \times 32768 \times 4096 \approx 4.40 \times 10^{12}.
> $$
>
> In contrast, when using the **linear** attention mechanism, the time complexity becomes
>
> $$
> ND^2 = 32768 \times 4096 \times 4096 \approx 5.50 \times 10^{11}.
> $$
>
> It can therefore be seen that the time complexity with linear attention is **one order of magnitude lower** than that with non-linear attention.
>
> Our statement that the time complexity of linear attention is $\mathcal{O}(N)$ is based on treating $D^2$ as a constant, with the core intention of conveying that the time complexity of linear attention grows linearly with respect to $N$. If you consider this formulation potentially ambiguous, we are happy to revise the wording accordingly. However, the conclusion that the time complexity of linear attention is lower than that of non-linear attention remains valid. A similar analysis can be found in Section 3.2.1 of [1].
>
> > [1] *Angelos Katharopoulos, et al. Transformers are RNNs: Fast Autoregressive Transformers with Linear Attention. ICML 2020.*

---

### Official Review · Reviewer_F58v · 2025-10-30

**Soundness:** 3
**Presentation:** 3
**Contribution:** 3
**Rating:** 6
**Confidence:** 5

**Summary:**

This paper addresses the critical issue of incorporating positional information into Spiking Transformers, as conventional methods from Artificial Neural Networks (ANNs) disrupt the essential spike-driven computational paradigm of Spiking Neural Networks (SNNs). The authors introduce a novel method called Spiking Positional Encoding (SPE), which is specifically designed to overcome this limitation. The core of SPE is the Positional Encoding Leaky Integrate-and-Fire (PE-LIF) neuron layer, which ingeniously encodes both absolute and relative positional information directly into the firing thresholds of neurons, thereby preserving the event-driven nature of SNNs. The authors provide a theoretical foundation for their method, proving that SPE can represent relative positions and exhibits a desirable long-term decay property. Through comprehensive experiments on seven datasets spanning time-series forecasting and natural language processing, SPE is shown to consistently outperform existing approaches and achieve state-of-the-art results, effectively bridging a significant performance gap between ANNs and SNNs in sequential modeling tasks.

**Strengths:**

This paper presents a compelling and well-executed study, exhibiting significant strengths across originality, quality, clarity, and significance. The originality of this work lies in its novel formulation of positional encoding specifically for Spiking Transformers. Instead of adapting ANN-based methods that disrupt the spike-driven paradigm, the authors introduce Spiking Positional Encoding (SPE), a creative solution that embeds positional information directly into the firing thresholds of a newly proposed PE-LIF neuron layer. This approach elegantly captures both absolute and relative positional information within a unified, spike-native framework. The quality of the research is exceptionally high, substantiated by both rigorous theoretical analysis and comprehensive empirical validation. The authors provide formal proofs (Propositions 1 and 2) to demonstrate SPE's capability to represent relative positions and its possession of a desirable long-term decay property. These theoretical claims are backed by extensive experiments across seven diverse datasets, where SPE consistently achieves state-of-the-art performance. Thorough ablation studies further strengthen the findings by methodically demonstrating the contribution of each component. The paper is presented with outstanding clarity; it logically progresses from a clear problem analysis to a set of well-defined design principles, which then guide the development of the proposed solution. The manuscript is well-written, and the high-quality figures effectively illustrate the core concepts. Finally, the work is highly significant as it addresses a critical bottleneck that has limited the application of SNNs to complex sequential tasks. By providing a principled and effective solution, this research substantially bridges the performance gap between ANNs and SNNs, advancing the field of neuromorphic computing and paving the way for the development of more powerful and energy-efficient, brain-inspired architectures.

**Weaknesses:**

1. The paper's persuasiveness is somewhat weakened by the lack of validation in the computer vision domain.
2. Hand-crafted, non-learnable positional thresholds. The PE-LIF thresholds adopt fixed sinusoidal formulas (Eq. 8), with even-dimension requirement and a global $\lambda$. This design may be under-adaptive across modalities, sequence scales, or layers, and the paper does not explore learnable or data-driven variants (e.g., per-layer amplitudes/phases or low-rank adapters).

**Questions:**

1. I would like to see your performance on static vision datasets as well as neuromorphic vision datasets.
2. Learnability of PE. Why not make the threshold modulation learnable (per-layer/per-head amplitudes, learnable frequencies/phases) and regularize toward sinusoidal priors? Would a learned variant outperform fixed sin/cos?
3. Where to place PE-LIF? Beyond the current placements (input/MLP tail and Q/K activations), did you evaluate (i) only Q, (ii) only K, (iii) V, or (iv) alternating layers? A placement study would clarify where positional signals are most beneficial.

---

> ### Author Response · Authors · 2025-11-26
> **Response to weakness 1 & question 1 (Part I)**
>
> We are deeply grateful for your time in reviewing our manuscript. Your insights will be of great help to us in improving the quality of our paper.
>
> > *Weakness 1: The paper's persuasiveness is somewhat weakened by the lack of validation in the computer vision domain.\
> > Question 1: I would like to see your performance on static vision datasets as well as neuromorphic vision datasets.*
>
> Response to weakness 1 & question 1: From the perspective of positional encoding, vision tasks differ fundamentally from sequential tasks such as time series modeling and NLP. In sequential tasks, positional information is one-dimensional, whereas in vision tasks it is intrinsically two-dimensional: the model must be informed of both the row and column indices of each feature. Consequently, the one-dimensional positional encoding proposed in our paper cannot be directly applied to vision models; otherwise, vertically adjacent patches may end up having very large distances in the relative positional encoding.
>
> We therefore extend the one-dimensional PE-LIF in the paper to a two-dimensional form:
> $$
> \theta_{x,y,k}=
> \begin{cases}
> \theta + \lambda\cdot \cos\left(\frac{x}{10000^{\frac{k-1}{D}}}\right) & x=1,2,...,N_x, k=1,5,...,D-3, \\newline
> \theta + \lambda\cdot \sin\left(\frac{x}{10000^{\frac{k-2}{D}}}\right) & x=1,2,...,N_x, k=2,6,...,D-2, \\newline
> \theta + \lambda\cdot \cos\left(\frac{y}{10000^{\frac{k-3}{D}}}\right) & y=1,2,...,N_y, k=3,7,...,D-1, \\newline
> \theta + \lambda\cdot \sin\left(\frac{y}{10000^{\frac{k-4}{D}}}\right) & y=1,2,...,N_y, k=4,8,...,D.
> \end{cases}
> $$
> Here, $x$ and $y$ denote the horizontal and vertical coordinates of the image, respectively. It can be shown that this two-dimensional PE-LIF is capable of representing two-dimensional relative positional information.
>
> > **Proposition 3** *When employing 2D version PE-LIF neuron layer to obtain activations*
> >
> > $$
>     \begin{cases}
>         Q(t)=[\boldsymbol{q} _{1,1}(t),\boldsymbol{q} _{1,2}(t),...,\boldsymbol{q} _{N_x,N_y}(t)]^\top \in \{0,1\}^{N_xN_y\times D},\\newline
>         K(t)=[\boldsymbol{k} _{1,1}(t),\boldsymbol{k} _{1,2}(t),...,\boldsymbol{k} _{N_x,N_y}(t)]^\top \in \{0,1\}^{N_xN_y\times D},
>     \end{cases}
> > $$
> >
> >*under the condition $\mathbb{E}\left[\hat{u}\_{x,y,k}(t)\right]=\mathbb{E}\left[s(t)\right]$ , there exists a function $g$ that takes $x_1-x_2$ and $y_1-y_2$ as its arguments such that $\mathbb{E}\left[\boldsymbol{q}^\top_{x1,y1}(t)\boldsymbol{k}_{x2,y2}(t)\right]$ contains $g(x_1-x_2,y_1-y_2)$, where*
> >
> > $$
> \begin{aligned}
> g(x_1-x_2,y_1-y_2)=&\frac{1}{2}\sum_{k=1}^{D/4}\left(\mathbb{E}\left[\mathcal{B}^{(x_1,x_2,y_1,y_2)} \_{4k-3,4k-3}\right]+\mathbb{E}\left[\mathcal{B}^{(x_1,x_2,y_1,y_2)} \_{4k-2,4k-2}\right]\right)\cos(x_1-x_2)\beta_k \\newline
> &+\frac{1}{2}\sum_{k=1}^{D/4}\left(\mathbb{E}\left[\mathcal{B}^{(x_1,x_2,y_1,y_2)}\_{4k-1,4k-1}\right]+\mathbb{E}\left[\mathcal{B}^{(x_1,x_2,y_1,y_2)}\_{4k,4k}\right]\right)\cos (y_1-y_2)\beta_k.
> \end{aligned}
> > $$
> >
> > *This demonstrates that 2D version PE-LIF can encode relative positional information directly into $Q(t)$ and $K(t)$.*

---

> ### Author Response · Authors · 2025-11-26
> **Response to weakness 1 & question 1 (Part II)**
>
> The proof of **Proposition 3** follows similar steps to that of **Proposition 1**. Below we give a detailed derivation for the parts where the differences are most pronounced:
> > **Proof**
> >
> > $$
> \begin{aligned}
> \mathbb{E}\left[\boldsymbol{\alpha} \_{x_1,y_1}^T\mathcal{B}^{(x_1,x_2,y_1,y_2)}\boldsymbol{\alpha} \_{x_2,y_2}\right]=&\sum_{k=1}^{D/4}\left(\mathbb{E}\left[\mathcal{B}^{(x_1,x_2,y_1,y_2)} \_{4k-3,4k-3}\right]\cos x_1\beta_k\cos x_2\beta_k+\mathbb{E}\left[\mathcal{B}^{(x_1,x_2,y_1,y_2)} \_{4k-2,4k-2}\right]\sin x_1\beta_k\sin x_2\beta_k\right)\\newline
>     &+\sum_{k=1}^{D/4}\left(\mathbb{E}\left[\mathcal{B}^{(x_1,x_2,y_1,y_2)} \_{4k-1,4k-1}\right]\cos y_1\beta_k\cos y_2\beta_k+\mathbb{E}\left[\mathcal{B}^{(x_1,x_2,y_1,y_2)} \_{4k,4k}\right]\sin y_1\beta_k\sin y_2\beta_k\right),\\newline
>     =&\frac{1}{2}\sum_{k=1}^{D/4}\left(\mathbb{E}\left[\mathcal{B}^{(x_1,x_2,y_1,y_2)} \_{4k-3,4k-3}\right]+\mathbb{E}\left[\mathcal{B}^{(x_1,x_2,y_1,y_2)} \_{4k-2,4k-2}\right]\right)\cos (x_1-x_2)\beta_k\\newline
>     &+\frac{1}{2}\sum_{k=1}^{D/4}\left(\mathbb{E}\left[\mathcal{B}^{(x_1,x_2,y_1,y_2)} \_{4k-3,4k-3}\right]-\mathbb{E}\left[\mathcal{B}^{(x_1,x_2,y_1,y_2)} \_{4k-2,4k-2}\right]\right)\cos (x_1+x_2)\beta_k\\newline
>     &+\frac{1}{2}\sum_{k=1}^{D/4}\left(\mathbb{E}\left[\mathcal{B}^{(x_1,x_2,y_1,y_2)} \_{4k-1,4k-1}\right]+\mathbb{E}\left[\mathcal{B}^{(x_1,x_2,y_1,y_2)} \_{4k,4k}\right]\right)\cos (y_1-y_2)\beta_k\\newline
>     &+\frac{1}{2}\sum_{k=1}^{D/4}\left(\mathbb{E}\left[\mathcal{B}^{(x_1,x_2,y_1,y_2)} \_{4k-1,4k-1}\right]-\mathbb{E}\left[\mathcal{B}^{(x_1,x_2,y_1,y_2)} \_{4k,4k}\right]\right)\cos (y_1+y_2)\beta_k,
> \end{aligned}
> > $$
> >
> > where $\beta_k=1/10000^{\frac{4(k-1)}{D}}$.
> Thus, we find that $g(x_1-x_2,y_1-y_2)$ is:
> >
> > $$
> \begin{aligned}
> g(x_1-x_2,y_1-y_2)=&\frac{1}{2}\sum_{k=1}^{D/4}\left(\mathbb{E}\left[\mathcal{B}^{(x_1,x_2,y_1,y_2)} \_{4k-3,4k-3}\right]+\mathbb{E}\left[\mathcal{B}^{(x_1,x_2,y_1,y_2)} \_{4k-2,4k-2}\right]\right)\cos(x_1-x_2)\beta_k \\newline
> &+\frac{1}{2}\sum_{k=1}^{D/4}\left(\mathbb{E}\left[\mathcal{B}^{(x_1,x_2,y_1,y_2)} \_{4k-1,4k-1}\right]+\mathbb{E}\left[\mathcal{B}^{(x_1,x_2,y_1,y_2)} \_{4k,4k}\right]\right)\cos (y_1-y_2)\beta_k.
> \end{aligned}
> > $$
> >
> > Therefore, **Proposition 3** is proved.
>
> We evaluate the 2D PE-LIF neuron on the ImageNet and DVS-CIFAR10 datasets, and the experimental results are summarized in the table below. The results demonstrate that our method can improve model performance. Due to the high computational cost of ImageNet, we conduct experiments only on the Spikingformer-8-256 model, while for DVS-CIFAR10 we adopt the Spikingformer-2-256 model.
>
> |Method|Acc.(DVS-CIFAR10)|Acc.(ImageNet)|
> |-|-|-|
> |Spikingformer|81.3|66.86|
> |Spikingformer w/ SPE|**81.5**|**67.59**|

---

> ### Author Response · Authors · 2025-11-26
> **Responses to weakness 2, 3 & question 2**
>
> > *Weakness 2: Hand-crafted, non-learnable positional thresholds. The PE-LIF thresholds adopt fixed sinusoidal formulas (Eq. 8), with even-dimension requirement and a global $\lambda$. This design may be under-adaptive across modalities, sequence scales, or layers, and the paper does not explore learnable or data-driven variants (e.g., per-layer amplitudes/phases or low-rank adapters).\
> > Question 2: Learnability of PE. Why not make the threshold modulation learnable (per-layer/per-head amplitudes, learnable frequencies/phases) and regularize toward sinusoidal priors? Would a learned variant outperform fixed sin/cos?*
>
> Response to weakness 2 & question 2: Your suggestion of introducing a learnable threshold is highly constructive. In our paper, we did not consider learnable thresholds, mainly because such thresholds might impair the ability of SPE to represent relative information (see **Proposition 1 & 2**). Combining your suggestion with our PE-LIF design, we propose a variant of PE-LIF in which the threshold is initialized using the fixed threshold of PE-LIF and then set to be learnable. We refer to this variant as **Learnable PE-LIF (LPE-LIF)**.
>
> We conducted experiments with LPE-LIF on the Electricity dataset, and the results are reported in the table below. For prediction lengths of 6, 48, and 96, PE-LIF with a non-learnable threshold performs better. However, for a prediction length of 24, LPE-LIF achieves better performance. This outcome may be related to the lower robustness of LPE-LIF.
>
> Overall, LPE-LIF demonstrates performance competitive with that of PE-LIF. Therefore, we will include LPE-LIF in a subsequent version of the paper as a variant of PE-LIF.
>
> |Method|R$^2_{PL=6}$|RSE$_{PL=6}$|R$^2_{PL=24}$|RSE$_{PL=24}$|R$^2_{PL=48}$|RSE$_{PL=48}$|R$^2_{PL=96}$|RSE$_{PL=96}$|
> |-|-|-|-|-|-|-|-|-|
> |PE-LIF|**0.983**|**0.233**|0.972|0.297|**0.972**|**0.297**|**0.964**|**0.337**|
> |**LPE-LIF**|0.956|0.373|**0.984**| **0.224**|0.960|0.357|0.956|0.373|
>
>
> > *Question 3: Where to place PE-LIF? Beyond the current placements (input/MLP tail and Q/K activations), did you evaluate (i) only Q, (ii) only K, (iii) V, or (iv) alternating layers? A placement study would clarify where positional signals are most beneficial.*
>
> Response to question 3:
> |Method|Only Q|Only K|Only V|Q & V|K & V|Q & K|
> |-|-|-|-|-|-|-|
> |R^$2$|0.958|0.964|0.970|0.932|0.947|**0.972**|
> |RSE|0.367|0.339|0.308|0.464|0.411|**0.297**|
>
> For the relative positional encoding, we evaluate six configurations: (i) applying it only to Q, (ii) only to K, (iii) only to V, (iv) to both Q and V, (v) to both K and V, and (vi) to both Q and K. The results show that when PE-LIF neurons are used to jointly activate Q and K, the model achieves the best performance. This demonstrates the effectiveness of our approach and supports the validity of **Proposition 1**. This experiment is conducted on the Electricity dataset with the prediction length set to 24.
>
> |Method|R$^2_{PL=6}$|RSE$_{PL=6}$|R$^2_{PL=24}$|RSE$_{PL=24}$|R$^2_{PL=48}$|RSE$_{PL=48}$|R$^2_{PL=96}$|RSE$_{PL=96}$|
> |-|-|-|-|-|-|-|-|-|
> |w/ MLP-APE|**0.983**|**0.233**|**0.972**|**0.297**|**0.972**|**0.297**|**0.964**|**0.337**|
> |w/o MLP-APE|0.948|0.403|0.948|0.406|0.958|0.363|0.946|0.414|
>
> For the absolute positional encoding, we compare our method with the conventional approach that adds absolute positional encodings only at the input layer, as shown in the table above. Here, “w/ MLP-APE” denotes the method proposed in this paper, while “w/o MLP-APE” denotes the variant that uses absolute positional encodings only at the input layer. The results indicate that w/ MLP-APE consistently outperforms w/o MLP-APE, which validates the effectiveness of our design. These experiments are conducted on the Electricity dataset. $PL$ denotes the prediction length.
>
> If you have any remaining concerns, please feel free to continue the discussion with us. Once again, I sincerely appreciate your valuable contributions to improving the quality of this paper.

---

> ### Comment · Reviewer_F58v · 2025-11-27
> **Response to the Rebuttal**
>
> I appreciate the authors’ comprehensive response, which has resolved my concerns. In view of the rebuttal, I find the motivation compelling and the experimental design robust; accordingly, I will raise my rating.

---

> > ### Author Response · Authors · 2025-11-27
> >
> > Dear reviewer F58v, we sincerely appreciate your positive evaluation of our work. We will carefully revise the manuscript in line with your suggestions, as well as those of the other reviewers, to further improve its quality.

---

### Official Review · Reviewer_QkMm · 2025-11-01

**Soundness:** 3
**Presentation:** 3
**Contribution:** 3
**Rating:** 4
**Confidence:** 5

**Summary:**

This paper proposes Spiking Positional Encoding (SPE) for spiking Transformers. The core mechanism replaces LIF layers with PE‑LIF layers whose firing thresholds are position-dependent, thereby embedding positional information into the spikes without breaking the spike‑driven computation paradigm. Absolute position is injected by using PE‑LIF in the first spike‑encoding layer and at the end of each MLP, while relative position is encoded by using PE‑LIF for the Q and K activations in Spiking Self‑Attention (SSA). They further introduce a membrane‑potential regularizer (MPR‑Loss) to approximately enforce the required expectation condition. Experiments on time‑series forecasting and text classification show consistent improvements.

**Strengths:**

1. SNN-compatible method. Encoding position via threshold modulation inside PE‑LIF is a clean, spike‑domain design that preserves binary spikes. This approach is simple, yet novel.
2. The proposed approach is supported by theoretical analysis.
3. MPR‑Loss operationalizes the expectation‑matching precondition and, in ablations on Solar with L=24, improves R² by about +0.022 and reduces RSE by 0.029. The accompanying histograms also show FR–mean alignment improvements in Q/K layers.
4. Well-written and organized manuscript.

**Weaknesses:**

1. The “longer‑horizon” advantage is not convincingly substantiated. The paper claims SPE is particularly effective for longer prediction horizons, but the deltas vs. the spike‑driven baseline are mixed. For example on Electricity, the R² gain of SPE over the spiking baseline is +0.027 at L=6 (0.983 vs. 0.956) and +0.021 at L=96 (0.964 vs. 0.943), which does not evidence a stronger effect at longer horizons; trends on other datasets should be similarly quantified, not only averaged.
2. Limited breadth of baselines on spiking Transformers with relative PE. The paper argues that relative PE usually breaks linear SSA; however, several linear‑attention–compatible relative/bias schemes (e.g., simple phase rotations on Q/K, ALiBi‑style biases) can be adapted without explicitly forming attention maps. A direct comparison (or at least a careful adaptation study) is missing, so it remains hard to isolate the value of encoding position in thresholds versus in Q/K phases or biases.
3. Where and how absolute PE is injected deviates from conventional approach and needs stronger justification. The method places APE at the very first spike‑encoding layer and at the end of each MLP. The rationale, and the effect of each insertion point are not fully analyzed.
4. Limited applications. The authors present experimental results applying their method to various tasks, but their application is limited compared to other related papers. How does the method perform on image classification benchmarks commonly used in ViT, such as ImageNet?
5. Energy analysis is absent.
6. Limited hyperparameter exploration and robustness anlaysis.
7. To verify that the proposed method works well as a PE, visualization of the proposed PE is required.

**Questions:**

Please refer to Weaknesses section.

---

> ### Author Response · Authors · 2025-11-26
> **Responses to weakness 1 & 2**
>
> We are deeply grateful for your time in reviewing our manuscript. Your insights will be of great help to us in improving the quality of our paper.
>
> > *Weakness 1: The “longer‑horizon” advantage is not convincingly substantiated. The paper claims SPE is particularly effective for longer prediction horizons, but the deltas vs. the spike‑driven baseline are mixed. For example on Electricity, the R² gain of SPE over the spiking baseline is +0.027 at L=6 (0.983 vs. 0.956) and +0.021 at L=96 (0.964 vs. 0.943), which does not evidence a stronger effect at longer horizons; trends on other datasets should be similarly quantified, not only averaged.*
>
> Response to weakness 1: The current description of the “longer-horizon” advantage in the paper lacks rigor, and we will revise this section in the next version of the manuscript.
>
> > *Weakness 2: Limited breadth of baselines on spiking Transformers with relative PE. The paper argues that relative PE usually breaks linear SSA; however, several linear‑attention–compatible relative/bias schemes (e.g., simple phase rotations on Q/K, ALiBi‑style biases) can be adapted without explicitly forming attention maps. A direct comparison (or at least a careful adaptation study) is missing, so it remains hard to isolate the value of encoding position in thresholds versus in Q/K phases or biases.*
>
> Response to weakness 2:
> > For *“limited breadth of baselines on spiking Transformers with relative PE.”*
>
> RoPE is a representative positional encoding method. In Table 1, we compare our method with directly applying RoPE to Spikformer, and our method achieves better performance.
>
> > For *“linear‑attention–compatible relative schemes (e.g., simple phase rotations on Q/K)”*
>
> RoPE is based on phase rotations applied to Q/K.
> **Although RoPE does not break linear SSA, directly applying RoPE to spiking Transformer models is not reasonable.** The formulation of RoPE is:
>
> $$ q_n^{rope} = \mathcal{R}_n q_n,\quad k_n^{rope} = \mathcal{R}_n k_n$$
>
> where $\mathcal{R}$ is a block-diagonal rotation matrix with the property:
>
> $$ \mathcal{R}_n^T \mathcal{R}_m = \mathcal{R} _{m-n} $$
>
> Therefore, RoPE is capable of representing relative positional information, since when computing the attention map we have:
>
> $$ q_n^{rope} \cdot k_m^{rope}= q_n^{T} \mathcal{R} _{n}^{T} \mathcal{R} _{m} k_m = q_n^T \mathcal{R} _{m-n} k_m$$
>
> and the matrix $\mathcal{R}_{m-n}$ encodes the relative position information.
> There are two possible ways to apply RoPE to spiking Transformer models. The first is to apply the rotation matrix $\mathcal{R}$ before the $q$ and $k$ vectors are activated by the LIF neurons. However, **this approach destroys the relative positional information**, because:
>
> $$ \text{LIF}(q_n^T\mathcal{R} _{n}^T)\text{LIF}(\mathcal{R} _{m} k_m) \neq \text{LIF}(q_n^T)\mathcal{R} _{m-n}\text{LIF}(k_m)$$
>
> The second way is to apply the rotation matrix $\mathcal{R}$ after $q$ and $k$ have been activated by the LIF neurons. Yet, **since $\mathcal{R}$ is a floating-point matrix, directly applying it to spike-form $q$ and $k$ violates the event-driven computation paradigm**.
> **In summary, directly applying RoPE to spiking Transformer models is not appropriate.**
>
> > For *“linear‑attention–compatible bias schemes (e.g., ALiBi‑style biases)”*
>
> A general formulation of linear attention is:
>
> $$ O=f(Q)h(K)^TV $$
>
> Linear attention first computes $h(K)^TV$, and under the assumption that the feature dimension $D$ is much smaller than the number of tokens $N$, it reduces the complexity of the attention mechanism from $\mathcal{O}(N^2)$ to $\mathcal{O}(N)$ (see Section 3.2 of the paper for a more detailed analysis). This is favorable for current **Large Language Models with extremely long token sequences**. In the SNN domain, the most commonly used attention mechanism, SSA, is also a form of linear attention.
>
> ALiBi-style positional encoding methods, represented by ALiBi, are characterized by the following core formulation:
> $$ \text{softmax}(QK^T+m\cdot B) $$
> Here, $B$ is a strictly lower-triangular matrix with linear bias, and when $i>j$, $B_{ij}=j-i$. Therefore, ALiBi must first compute $QK^T$, i.e., the attention map, and only then can the matrix $B$ encoding relative positional information be added. **This makes ALiBi incompatible with linear attention mechanisms.**

---

> ### Author Response · Authors · 2025-11-26
> **Response to weakness 3**
>
> > *Weakness 3: Where and how absolute PE is injected deviates from conventional approach and needs stronger justification. The method places APE at the very first spike‑encoding layer and at the end of each MLP. The rationale, and the effect of each insertion point are not fully analyzed.*
>
> Response to weakness 3:
> |Method|R$^2_{PL=6}$|RSE$_{PL=6}$|R$^2_{PL=24}$|RSE$_{PL=24}$|R$^2_{PL=48}$|RSE$_{PL=48}$|R$^2_{PL=96}$|RSE$_{PL=96}$|
> |-|-|-|-|-|-|-|-|-|
> |w/ MLP-APE|**0.983**|**0.233**|**0.972**|**0.297**|**0.972**|**0.297**|**0.964**|**0.337**|
> |w/o MLP-APE|0.948|0.403|0.948|0.406|0.958|0.363|0.946|0.414|
>
> Mainstream absolute positional encoding methods, such as the sinusoidal positional encoding in Transformer [1] and the learnable positional encoding in BERT [2], are typically added to the token embeddings before being fed into the model. This is analogous to how we introduce APE in the very first spike-encoding layer. In contrast, instead of injecting absolute positional information only at the model input, we add it to the input of every attention layer. Our motivation is that, if absolute positional encodings are introduced only at the input layer, their positional information will be gradually “diluted” by subsequent transformations as the network becomes deeper, thereby weakening their impact on higher-level representations. To ensure that each self-attention layer can directly access sufficient absolute positional information, we repeatedly inject the absolute positional encodings at the input of all attention layers.
>
> We compare our method with the conventional approach that adds absolute positional encodings only at the model input, and the results are shown in the table above. Here, “w/ MLP-APE” denotes the method proposed in our paper, while “w/o MLP-APE” denotes the variant that only applies absolute positional encodings at the model input. The results show that “w/ MLP-APE” achieves better performance, which substantiates the soundness of our design.
> The above experiments are conducted on the Electricity dataset. In the table above, (PL) denotes the prediction length.
>
> > [1] *Ashish Vaswani, et al. Attention Is All You Need. NeurIPS 2017.*\
> > [2] *Devlin Jacob, et al. Bert: Pre-training of deep bidirectional transformers for language understanding. NAACL 2019.*

---

> ### Author Response · Authors · 2025-11-26
> **Response to weakness 4 (Part I)**
>
> > *Weakness 4: Limited applications. The authors present experimental results applying their method to various tasks, but their application is limited compared to other related papers. How does the method perform on image classification benchmarks commonly used in ViT, such as ImageNet?*
>
> Response to weakness 4: From the perspective of positional encoding, vision tasks differ fundamentally from sequential tasks such as time series modeling and NLP. In sequential tasks, positional information is one-dimensional, whereas in vision tasks it is intrinsically two-dimensional: the model must be informed of both the row and column indices of each feature. Consequently, the one-dimensional positional encoding proposed in our paper cannot be directly applied to vision models; otherwise, vertically adjacent patches may end up having very large distances in the relative positional encoding.
>
> We therefore extend the one-dimensional PE-LIF in the paper to a two-dimensional form:
> $$
> \theta_{x,y,k}=
> \begin{cases}
> \theta + \lambda\cdot \cos\left(\frac{x}{10000^{\frac{k-1}{D}}}\right) & x=1,2,...,N_x, k=1,5,...,D-3, \\newline
> \theta + \lambda\cdot \sin\left(\frac{x}{10000^{\frac{k-2}{D}}}\right) & x=1,2,...,N_x, k=2,6,...,D-2, \\newline
> \theta + \lambda\cdot \cos\left(\frac{y}{10000^{\frac{k-3}{D}}}\right) & y=1,2,...,N_y, k=3,7,...,D-1, \\newline
> \theta + \lambda\cdot \sin\left(\frac{y}{10000^{\frac{k-4}{D}}}\right) & y=1,2,...,N_y, k=4,8,...,D.
> \end{cases}
> $$
> Here, $x$ and $y$ denote the horizontal and vertical coordinates of the image, respectively. It can be shown that this two-dimensional PE-LIF is capable of representing two-dimensional relative positional information.
>
> > **Proposition 3** *When employing 2D version PE-LIF neuron layer to obtain activations*
> >
> > $$
>     \begin{cases}
>         Q(t)=[\boldsymbol{q} _{1,1}(t),\boldsymbol{q} _{1,2}(t),...,\boldsymbol{q} _{N_x,N_y}(t)]^\top \in \{0,1\}^{N_xN_y\times D},\\newline
>         K(t)=[\boldsymbol{k} _{1,1}(t),\boldsymbol{k} _{1,2}(t),...,\boldsymbol{k} _{N_x,N_y}(t)]^\top \in \{0,1\}^{N_xN_y\times D},
>     \end{cases}
> > $$
> >
> >*under the condition $\mathbb{E}\left[\hat{u}\_{x,y,k}(t)\right]=\mathbb{E}\left[s(t)\right]$ , there exists a function $g$ that takes $x_1-x_2$ and $y_1-y_2$ as its arguments such that $\mathbb{E}\left[\boldsymbol{q}^\top_{x1,y1}(t)\boldsymbol{k}_{x2,y2}(t)\right]$ contains $g(x_1-x_2,y_1-y_2)$, where*
> >
> > $$
> \begin{aligned}
> g(x_1-x_2,y_1-y_2)=&\frac{1}{2}\sum_{k=1}^{D/4}\left(\mathbb{E}\left[\mathcal{B}^{(x_1,x_2,y_1,y_2)} \_{4k-3,4k-3}\right]+\mathbb{E}\left[\mathcal{B}^{(x_1,x_2,y_1,y_2)} \_{4k-2,4k-2}\right]\right)\cos(x_1-x_2)\beta_k \\newline
> &+\frac{1}{2}\sum_{k=1}^{D/4}\left(\mathbb{E}\left[\mathcal{B}^{(x_1,x_2,y_1,y_2)}\_{4k-1,4k-1}\right]+\mathbb{E}\left[\mathcal{B}^{(x_1,x_2,y_1,y_2)}\_{4k,4k}\right]\right)\cos (y_1-y_2)\beta_k.
> \end{aligned}
> > $$
> >
> > *This demonstrates that 2D version PE-LIF can encode relative positional information directly into $Q(t)$ and $K(t)$.*

---

> ### Author Response · Authors · 2025-11-26
> **Response to weakness 4 (Part II)**
>
> The proof of **Proposition 3** follows similar steps to that of **Proposition 1**. Below we give a detailed derivation for the parts where the differences are most pronounced:
> > **Proof**
> >
> > $$
> \begin{aligned}
> \mathbb{E}\left[\boldsymbol{\alpha} \_{x_1,y_1}^T\mathcal{B}^{(x_1,x_2,y_1,y_2)}\boldsymbol{\alpha} \_{x_2,y_2}\right]=&\sum_{k=1}^{D/4}\left(\mathbb{E}\left[\mathcal{B}^{(x_1,x_2,y_1,y_2)} \_{4k-3,4k-3}\right]\cos x_1\beta_k\cos x_2\beta_k+\mathbb{E}\left[\mathcal{B}^{(x_1,x_2,y_1,y_2)} \_{4k-2,4k-2}\right]\sin x_1\beta_k\sin x_2\beta_k\right)\\newline
>     &+\sum_{k=1}^{D/4}\left(\mathbb{E}\left[\mathcal{B}^{(x_1,x_2,y_1,y_2)} \_{4k-1,4k-1}\right]\cos y_1\beta_k\cos y_2\beta_k+\mathbb{E}\left[\mathcal{B}^{(x_1,x_2,y_1,y_2)} \_{4k,4k}\right]\sin y_1\beta_k\sin y_2\beta_k\right),\\newline
>     =&\frac{1}{2}\sum_{k=1}^{D/4}\left(\mathbb{E}\left[\mathcal{B}^{(x_1,x_2,y_1,y_2)} \_{4k-3,4k-3}\right]+\mathbb{E}\left[\mathcal{B}^{(x_1,x_2,y_1,y_2)} \_{4k-2,4k-2}\right]\right)\cos (x_1-x_2)\beta_k\\newline
>     &+\frac{1}{2}\sum_{k=1}^{D/4}\left(\mathbb{E}\left[\mathcal{B}^{(x_1,x_2,y_1,y_2)} \_{4k-3,4k-3}\right]-\mathbb{E}\left[\mathcal{B}^{(x_1,x_2,y_1,y_2)} \_{4k-2,4k-2}\right]\right)\cos (x_1+x_2)\beta_k\\newline
>     &+\frac{1}{2}\sum_{k=1}^{D/4}\left(\mathbb{E}\left[\mathcal{B}^{(x_1,x_2,y_1,y_2)} \_{4k-1,4k-1}\right]+\mathbb{E}\left[\mathcal{B}^{(x_1,x_2,y_1,y_2)} \_{4k,4k}\right]\right)\cos (y_1-y_2)\beta_k\\newline
>     &+\frac{1}{2}\sum_{k=1}^{D/4}\left(\mathbb{E}\left[\mathcal{B}^{(x_1,x_2,y_1,y_2)} \_{4k-1,4k-1}\right]-\mathbb{E}\left[\mathcal{B}^{(x_1,x_2,y_1,y_2)} \_{4k,4k}\right]\right)\cos (y_1+y_2)\beta_k,
> \end{aligned}
> > $$
> >
> > where $\beta_k=1/10000^{\frac{4(k-1)}{D}}$.
> Thus, we find that $g(x_1-x_2,y_1-y_2)$ is:
> >
> > $$
> \begin{aligned}
> g(x_1-x_2,y_1-y_2)=&\frac{1}{2}\sum_{k=1}^{D/4}\left(\mathbb{E}\left[\mathcal{B}^{(x_1,x_2,y_1,y_2)} \_{4k-3,4k-3}\right]+\mathbb{E}\left[\mathcal{B}^{(x_1,x_2,y_1,y_2)} \_{4k-2,4k-2}\right]\right)\cos(x_1-x_2)\beta_k \\newline
> &+\frac{1}{2}\sum_{k=1}^{D/4}\left(\mathbb{E}\left[\mathcal{B}^{(x_1,x_2,y_1,y_2)} \_{4k-1,4k-1}\right]+\mathbb{E}\left[\mathcal{B}^{(x_1,x_2,y_1,y_2)} \_{4k,4k}\right]\right)\cos (y_1-y_2)\beta_k.
> \end{aligned}
> > $$
> >
> > Therefore, **Proposition 3** is proved.
>
> We apply the above 2D PE-LIF neurons to vision tasks on the ImageNet dataset, with the corresponding results summarized in the table below. The results show that our method improves model performance. Due to the high computational cost of ImageNet, we only conduct experiments on the Spikingformer-8-256 model.
>
> |Method|Acc.|
> |-|-|
> |Spikingformer|66.86|
> |Spikingformer w/ SPE|**67.59**|

---

> ### Author Response · Authors · 2025-11-26
> **Responses to weakness 5, 6 & 7**
>
> > *Weakness 5: Energy analysis is absent.*
>
> Response to weakness 5: The primary difference between PE-LIF and LIF neurons lies in their firing thresholds. Assuming similar input distributions, different thresholds will lead to different firing rates $fr$ for the two types of neurons. The network architectures using these two neuron models are identical, and thus their $FLOPs$ are the same. Since
> $$SOP^l=fr\times T\times FLOPs^l,$$
> and
> $$E_{SNN}=E_{AC}\times(\sum_{i=2}^N SOP^i_{Conv}+\sum_{j=1}^MSOP^j_{SSA})+E_{MAC}\times FLOPs^1_{Conv},$$
> it is sufficient to compare the firing rates $fr$ of the two neuron types in order to compare their energy consumption $E_{SNN}$. The table below reports the firing rates of PE-LIF and LIF neurons on the ETT test set. Except for the first layer, where PE-LIF exhibits a slightly higher firing rate, **PE-LIF consistently shows significantly lower firing rates than LIF in all subsequent layers, indicating that PE-LIF consumes less energy than LIF**.
>
> |Layer|PE-LIF|LIF|
> |-|-|-|
> |Init neuron layer|0.008|**0.002**|
> |Layer1.Q|**0.062**|0.088|
> |Layer1.K|**0.028**|0.110|
> |Layer1.MLP|**0.023**|0.127|
> |Layer2.Q|**0.046**|0.100|
> |Layer2.K|**0.023**|0.071|
> |Layer2.MLP|**0.041**|0.084|
>
> > *Weakness 6: Limited hyperparameter exploration and robustness anlaysis.*
>
> Response to weakness 6:
> |$\lambda$|0.1|0.15|0.2|0.25|0.3|0.35|0.4|
> |-|-|-|-|-|-|-|-|
> |R$^2$|0.939|0.962|0.952|0.958|**0.972**|0.968|0.967|
> |RSE|0.441|0.347|0.388|0.367|**0.297**|0.318|0.324|
>
> The main hyperparameter of the PE-LIF method is the $\lambda$ in Eq. (8), which is set to $0.3$ in our experiments. On the Electricity dataset with $L=24$, we conduct a comparison over different values of $\lambda$, and the results are reported in the table above. The model achieves the best performance when $\lambda=0.3$; its performance remains high for $\lambda \in \{0.3,0.35,0.4\}$, but degrades for other values, indicating that the method is relatively robust to the choice of $\lambda$. Since we expect the threshold to stay close to $1$, excessively large values of $\lambda$ are undesirable.
>
> > *Weakness 7: To verify that the proposed method works well as a PE, visualization of the proposed PE is required.*
>
> Response to weakness 7: Since positional information cannot be directly disentangled and visualized from the feature maps obtained under standard inference, we instead feed a constant value into PE-LIF over multiple time steps and visualize the resulting feature maps to isolate the positional information. As images cannot be uploaded in the rebuttal, we will include the visualizations of the positional encodings, together with the corresponding analysis, in a subsequent version of the paper.
>
> If you have any remaining concerns, please feel free to continue the discussion with us. Once again, I sincerely appreciate your valuable contributions to improving the quality of this paper.

---

> ### Author Response · Authors · 2025-11-27
>
> Dear Reviewer QkMm,
>
> Thank you very much for your time in reviewing our manuscript and for your valuable comments. As the author–reviewer discussion phase is drawing to a close, we would like to confirm whether our responses have satisfactorily addressed your concerns. We have provided detailed point-by-point replies, and we hope they resolve the issues you raised. Should you require any further clarification or have additional questions, please do not hesitate to let us know. We would be pleased to continue the discussion.
>
> Best regards.

---

### Meta-Review · Area_Chair_Xs3Z · 2025-12-27

**Summary:**

This paper introduces a new technique to handle positional information in Spiking Transformers that use brain-like "spikes" instead of regular MAC operations. Normal AI models use schemes like sinusoidal positional encodings (PE), RoPE, or ALiBi to understand word order or sequence position, but these do NOT work well in spiking models because everything runs on spikes.

The authors propose Spiking Positional Encoding (SPE), which uses a special neuron called PE-LIF. It changes how neurons fire by adjusting their "firing thresholds" to encode both absolute position (in the sequence) and relative position (how far apart two are). They include theoretic analysis to support and test it on time-series forecasting, language tasks, and some vision tasks.

The paper feels overly complicated, focused on a narrow problem, and hard to judge for real newness or impact. Reviewers were split: some liked the theory and effort, others were skeptical about whether it is truly novel.

**Reviewer Concerns:**

Main Concerns Shared Reviewers

1. Is This New, or Old Ideas in a Spiking Outfit? Many reviewers wondered if SPE is basically the same as RoPE (a popular positional encoding) but rewritten for spikes by tweaking neuron thresholds. The authors say RoPE cannot be used directly in spiking models, but the new method still looks like sinusoidal patterns — just moved into the spiking rules. **So this paper is more like an adaptation than a groundbreaking new concept.**

2. Big Claims, Small Results. The authors claimed SPE "bridges the gap between regular AI and spiking AI" and is especially good for long-term predictions. But the actual improvements over other spiking methods are usually small. The "long-horizon advantage" was admitted to not be fully proven in the first version. Some gains seem to work only on certain datasets, not across the board. There is a gap between the exciting promises.

3. Tests Are Limited. The authors tried many datasets, but: (1) Models are often small or very specific. (2) Comparisons to strong regular (non-spiking) AI baselines are weak or missing. (3) Vision tests (like ImageNet) were added late, use only one setup, and are very limited.
Therefore, it is diffcult to know if this would make a real difference in big, practical spiking Transformers.

4. Complexity and Energy Claims Are Messy. The authors discussed about how fast the method is (claiming O(N) complexity for attention), but at least one reviewer called this a serious mistake. The authors later said they treat some terms as constant, but it still feels sloppy for a paper that emphasizes theory. Energy savings (important for spiking hardware) are based on rough firing-rate guesses, not real full measurements — so those claims feel weak.

5. Lots of Hand-Tuning, Little Solid Reasoning. The method uses fixed (non-learnable) patterns, global settings, and specific places to insert the positional information. At first, there was little explanation why these choices. Even after adding tests and a learnable version, it feels like the method was heavily tweaked to work, not naturally robust.

Reviewer-Specific Questions:

1. Several reviewers (QkMm, SaAm, 6QkJ) had serious technical issues: not enough newness, missing detailed tests, confusing math on complexity, small models, and no real energy checks.

2. One (SaAm) kept doubting the complexity claims even after the authors explained — suggesting deeper problems.

More reviewers rated this paper as "4: marginally below the acceptance threshold."

**Reviewer Scores:**

The reviewers' scores are reasonable.

---

### Decision · Program_Chairs · 2026-01-26

Reject